# Simplified and Generalized Masked Diffusion for Discrete Data

**Jiaxin Shi**\*, **Kehang Han**\*, **Zhe Wang, Arnaud Doucet, Michalis K. Titsias**
Google DeepMind

## Abstract

Masked (or absorbing) diffusion is actively explored as an alternative to autoregressive models for generative modeling of discrete data. However, existing work in this area has been hindered by unnecessarily complex model formulations and unclear relationships between different perspectives, leading to suboptimal parameterization, training objectives, and ad hoc adjustments to counteract these issues. In this work, we aim to provide a simple and general framework that unlocks the full potential of masked diffusion models. We show that the continuous-time variational objective of masked diffusion models is a simple weighted integral of cross-entropy losses. Our framework also enables training generalized masked diffusion models with state-dependent masking schedules. When evaluated by perplexity, our models trained on OpenWebText surpass prior diffusion language models at GPT-2 scale and demonstrate superior performance on 4 out of 5 zero-shot language modeling tasks. Furthermore, our models vastly outperform previous discrete diffusion models on pixel-level image modeling, achieving 2.75 (CIFAR-10) and 3.40 (ImageNet $64{\times}64$) bits per dimension that are better than autoregressive models of similar sizes. Our code is available at https://github.com/google-deepmind/md4.

## 1 Introduction

Since their inception [1, 2, 3], diffusion models have emerged as the workhorse for generative media, achieving state-of-the-art in tasks such as image synthesis [4, 5, 6], audio [7, 8] and video generation [9, 10, 11, 12, 13]. The majority of existing successes are for continuous state space diffusions. While diffusion models have been extended to discrete state spaces [1, 14, 15] and have been successfully applied to applications ranging from graph generation [16], text-to-sound generation [17] or protein design [18], they remain not as widely used as their continuous counterparts as they are not competitive with autoregressive models in important domains such as text modeling. This has motivated the development of continuous space diffusion models where the discrete data are embedded in the Euclidean space [19, 20, 21, 22, 23] or the simplex [24, 25, 26, 27, 28]. We believe that one of the reasons for the limited success of discrete diffusions is that they have been hindered by fairly complex formulations and training objectives. This paper is a step towards closing this gap.

In this work, we focus on "masked" (or "absorbing") diffusions, a discrete diffusion formulation first presented by Austin et al. [14], and later explored by the literature from various perspectives [29, 30, 31, 32]. We follow here a continuous-time framework which has proven very useful to improve the training and understanding of continuous state space diffusions [see e.g., 3, 33, 34]. We make several technical contributions which simplify the training of these models and improve significantly their performance. Our contributions are as follows:

- Using elementary arguments, we establish several properties for the forward process induced by this model and its corresponding time reversal, improving our understanding of this model class.

---

\*Equal contribution. Correspondence to: jiaxins@google.com.

38th Conference on Neural Information Processing Systems (NeurIPS 2024).

- We provide a remarkably simple expression of the Evidence Lower Bound (ELBO) for masked diffusion models, showing that it corresponds to a weighted integral over time of cross-entropy losses. Similarly to continuous space diffusions [33], this objective can be rewritten in terms of signal-to-noise ratio and exhibits invariance properties.

- We develop a unifying understanding of previously proposed continuous-time discrete diffusion models [29, 32, 35], revealing the changes they made to our ELBO objective and/or model parameterization. We show that these changes either lead to expensive model evaluations, or large variance in training, or breaking the consistency between forward and reverse processes.

- On GPT-2 scale text modeling and pixel-level image modeling tasks, masked diffusions trained using our simple ELBO objective outperform previous proposals, leading to the best likelihood and zero-shot transfer performance among discrete diffusion models.

- Finally, based on our simplified masked diffusion formulation, we propose a generalized masked diffusion model that allows state-dependent masking schedules. This generalized masked diffusion model further improves predictive performance measured by test likelihoods.

Concurrent work by Ou et al. [36] and Sahoo et al. [37] derives a similar simplified expression of the ELBO. Ou et al. [36]'s derivation relies on an observation similar to the one we made in Proposition 1.

## 2 Masked Diffusion

Consider a sentence where we progressively replace each word with a special mask token, transforming the sentence into a sequence of masks. Our goal is to train a generative model that reverses this process, effectively turning a sentence of masks back into meaningful text. More formally, assume our data consists of tokens from a finite discrete state space with $m$ possible states, represented by integers $0, 1, \ldots, m-1$ and their corresponding one-hot vectors $e_0, e_1, \ldots, e_{m-1}$. To accommodate the masking process, we augment this space with an additional mask state, denoted by the index $m$. The masking process transitions each token to the mask state at a random time. This process, known as the forward process, is applied independently to each token (e.g., each word), progressively converting the data into a sequence of mask tokens. By learning to reverse this masking process, we create a generative model capable of producing coherent discrete data.

**Discrete-time forward process.** We start with the case of a single token and later expand to multiple dimensions. We define the forward process as a Markovian sequence of discrete random variables $x_t$ indexed by time $t$, where $t$ runs from 0 to 1. Throughout the work, we abuse the notation such that $x_t$ can be either an integer or its corresponding one-hot vector, whenever it is clear from the context. We divide $[0, 1]$ into $T$ intervals, and let $s(i) = (i-1)/T$, $t(i) = i/T$. Following Austin et al. [14], the state transition between $[s(i), t(i)]$ is determined by a transition matrix of size $(m+1) \times (m+1)$: $Q_i = (1 - \beta_i)I + \beta_i \mathbf{1} e_m^\top$, where $\mathbf{1}$ is an all-one vector of size $m+1$, $e_m$ represents a one-hot vector where element at index $m$ is 1. Each entry $[Q_i]_{jk}$ denotes the probability of transition from the state $j$ to the state $k$:

$$[Q_i]_{jk} = q(x_{t(i)} = k | x_{s(i)} = j) = (1 - \beta_i)\delta_{jk} + \beta_i \delta_{km}.$$

This means that, with probability $1 - \beta_i$, $x_{t(i)} = x_{s(i)}$, otherwise it jumps to the mask state. Given the above transition matrix, the marginal distribution at time $t(i)$ given $x_0$ is

$$q(x_{t(i)} | x_0) = \text{Cat}(x_{t(i)}; \bar{Q}_i^\top x_0) = x_0^\top \bar{Q}_i x_{t(i)}.$$

Here, we use $\text{Cat}(x; p)$ to denote a Categorical distribution where $p$ is the vector of probabilities of being in each category, and $\bar{Q}_i \triangleq \prod_{j=1}^i Q_j = \alpha_i I + (1 - \alpha_i)\mathbf{1} e_m^\top$ for $\alpha_i = \prod_{j=1}^i (1 - \beta_j)$. We expect $\alpha_T$ to become very small or zero for a sufficiently large $T$ such that $q(x_1 | x_0)$ for any $x_0$ will become a delta mass at the mask state.

**Continuous-time limit.** We can define a continuous-time forward process by taking a limit of the above discrete-time process. We first specify a continuous function $\beta(t)$ such that $\beta_i = \beta(t(i))/T$. We then let $T \to \infty$ in the discrete-time process and compute the limit of $\bar{Q}_i$ (proved in Austin et al. 14, Appendix A.6, see also App. A) as

$$\bar{Q}(t) \triangleq \lim_{T \to \infty} \bar{Q}_i = \alpha_t I + (1 - \alpha_t)\mathbf{1} e_m^\top, \text{ where } \alpha_t \triangleq \exp\left(-\int_0^t \beta(s)\mathrm{d}s\right), \tag{1}$$

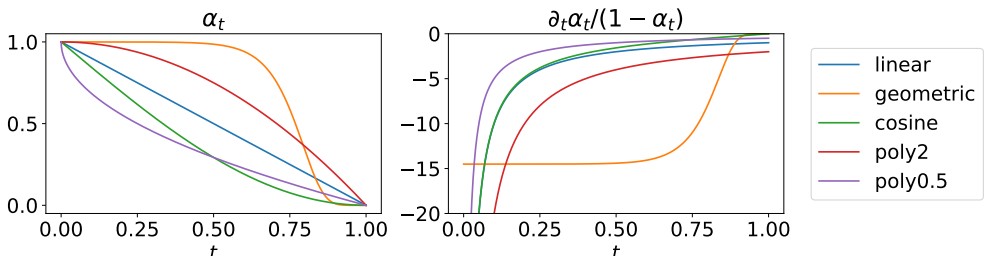

Figure 1: Masking schedules in the literature: (Left) $\alpha_t$; (Right) weight of the cross-entropy loss w.r.t. $t$; Equations for these schedules are given in Tab. 4 in Appendix.

so that $q(x_t|x_0) = \text{Cat}(x_t; \bar{Q}(t)^\top x_0)$. For two arbitrary times, $0 \leq s < t \leq 1$, the transition distribution that is compatible with the above marginal (i.e., $q(x_t|x_0) = \sum_{x_s} q(x_t|x_s)q(x_s|x_0)$) is

$$q(x_t|x_s) = \text{Cat}(x_t; \bar{Q}(s,t)^\top x_s), \text{ where } \bar{Q}(s,t) \triangleq \bar{Q}(s)^{-1}\bar{Q}(t) = \frac{\alpha_t}{\alpha_s}I + \left(1 - \frac{\alpha_t}{\alpha_s}\right)\mathbf{1}e_m^\top.$$

Note that Austin et al. [14] did not derive this explicit form of transition matrix between two arbitrary time $s$ and $t$, which appeared later in Zhao et al. [38] concurrently with our work.

**Masking schedules.** From the definition of $\alpha_t$, we have that $\alpha_0 = 1$. And similar to the discrete-time formulation, we would like $\alpha_1$ be zero or very close to zero. We provide a summary of masking schedules from literature that satisfy these properties in Fig. 1. The linear schedule was proposed in Sohl-Dickstein et al. [1] for binary variables and then re-derived by Austin et al. [14] from mutual information for discrete-time models. The geometric schedule $\alpha_t$ is plotted for $\bar{\beta}_{\min} = 10^{-5}$ and $\bar{\beta}_{\max} = 20$. It was first used for continuous diffusions [3] and then for discrete by Lou et al. [32]. The cosine schedule was originally proposed in MaskGIT [39], an iterative unmasking generative model inspired by diffusion. This schedule has the property of slowing down the unmasking process at the beginning of the reverse generation. Aligning with their observation, we find that this results in a lower chance of conflicting tokens being unmasked simultaneously at the start of generation, thereby enhancing the overall generation quality.

**Time reversal of the forward process given $x_0$.** The analytic property of our forward process allows to compute many quantities of interest in closed form. One such quantity frequently used in diffusion models is the time reversal of the forward process given $x_0$: $q(x_s|x_t, x_0)$ for $s \leq t$. We derive it in App. C as

$$q(x_s|x_t, x_0) = \text{Cat}(x_s; \bar{R}^{x_0}(t,s)^\top x_t), \text{ where } \bar{R}^{x_0}(t,s) = I + \frac{\alpha_s - \alpha_t}{1 - \alpha_t}e_m(x_0 - e_m)^\top.$$

From the transition matrix $\bar{R}^{x_0}(t,s) \in \mathbb{R}^{(m+1)\times(m+1)}$ we can see the reverse process conditioned on $x_0$ has a very simple logic—if $x_t$ is a mask, with probability $\frac{\alpha_s - \alpha_t}{1 - \alpha_t}$, it will jump to the state $x_0$ at time $s$, otherwise it will stay masked. Once $x_t$ is unmasked, it remains in the same state until the end.

## 3 Model and Objective

For a discrete-time masked diffusion process, we define our generative model by approximately reversing the forward transitions using a reverse model $p_\theta(x_s|x_t)$. One way to define this model is

$$p_\theta(x_s|x_t) \triangleq q(x_s|x_t, \mu_\theta(x_t, t)), \tag{2}$$

where $\mu_\theta(x_t, t) \in \Delta^{m+1}$ is a probability vector parametrized by a neural network $f_\theta$ with a softmax applied to the output logits (note the $m$-th output is forced to 0 since the clean data cannot be masks):

$$\mu_\theta(x_t, t) = \begin{cases} \text{softmax}(f_\theta(x_t, t)) & x_t = m, \\ x_t & x_t \neq m. \end{cases} \tag{3}$$

This is known as mean-parameterization since it leverages a prediction model for the mean of $x_0$. A matrix-form depiction of $p_\theta(x_s|x_t)$ is shown in Fig. 7 (right). In fact, we can select a time-invariant parametrization $\mu_\theta(x_t, t) = \mu_\theta(x_t)$ as [36] showed that $p(x_0|x_t)$ given $x_t = x$ is identical for any $t$.

Besides $p_\theta(x_s|x_t)$, we also need to specify $p(x_0|x_{t(1)})$ and the prior distribution $p(x_{t(T)}) = p(x_1)$. Following the practice in continuous diffusion models [33], we choose $p(x_0|x_{t(1)}) \propto q(x_{t(1)}|x_0)$. And since $q(x_1|x_0) \approx \delta_{x_1,m}$ for any $x_0$ as $\alpha_1 \approx 0$, we set $p(x_1) \approx \delta_{x_1,m}$, see App. E.

We then write out the discrete-time diffusion model objective [1, 2], which is a lower bound of the log marginal likelihood of data $x_0$ under the model $p$ (known as the Evidence Lower Bound, or ELBO):

$$\log p(x_0) \geq \mathbb{E}_{q(x_{t(1)}|x_0)}[\log p(x_0|x_{t(1)})] - \mathrm{KL}(q(x_1|x_0)\|p(x_1)) - \mathcal{L}_T,$$

where $\mathcal{L}_T = \sum_{i=2}^{T} \mathbb{E}_{q(x_{t(i)}|x_0)}[\mathrm{KL}(q(x_{s(i)}|x_{t(i)}, x_0)\|p_\theta(x_{s(i)}|x_{t(i)}))]$. For the above choices of the prior distribution, the term $\mathrm{KL}(q(x_1|x_0)\|p(x_1))$ becomes zero. Under the reverse model (2), the KL divergence terms in $\mathcal{L}_T$ becomes (proof in App. D)

$$\mathrm{KL}(q(x_s|x_t, x_0)\|p_\theta(x_s|x_t)) = -\frac{\alpha_s - \alpha_t}{1 - \alpha_t}\delta_{x_t,m} \cdot x_0^\top \log \mu_\theta(x_t, t),$$

which is a simple cross-entropy loss between the predicted logits and the clean data. In App. D, we show that $\mathcal{L}_T$ is a Riemann sum and is lower bounded by the corresponding continuous integral:

$$\mathcal{L}_\infty \triangleq \lim_{T \to \infty} \mathcal{L}_T = \int_{t(1)}^{1} \frac{\alpha_t'}{1 - \alpha_t}\mathbb{E}_{q(x_t|x_0)}\left[\delta_{x_t,m} \cdot x_0^\top \log \mu_\theta(x_t, t)\right] \mathrm{d}t, \tag{4}$$

where $\alpha_t'$ denotes the derivative of $\alpha_t$ with respect to $t$. Therefore, we can obtain an ELBO that is tighter than that of any finite $T$ by pushing $T \to \infty$. This ELBO can be further simplified by letting $t(1) \to 0$. As a result, $\mathbb{E}_{q(x_{t(1)}|x_0)}[\log p(x_0|x_{t(1)})]$ goes to 0 and the ELBO becomes $-\mathcal{L}_\infty$.

For continuous state-space diffusions, the ELBO depends on the signal-to-noise ratio (SNR) at its endpoints but is otherwise invariant to the noise schedule [33]. We establish here a similar result for discrete diffusions. Consider choosing $\alpha_t = \sigma(\lambda_t)$, where $\sigma$ represents the sigmoid function $\sigma(x) = \frac{1}{1+e^{-x}}$. In this context, the log-SNR is defined by $\lambda_t = \log\frac{\alpha_t}{1-\alpha_t} = \log\text{-SNR}(t)$. By making a change of variables in (4) to make everything a function of the log-SNR, we obtain

$$\mathcal{L}_\infty = \int_{\lambda_{t(1)}}^{\lambda_1} \sigma(\lambda)\mathbb{E}_{\tilde{q}(x_\lambda|x_0)}\left[\delta_{x_\lambda,m} \cdot x_0^\top \log \tilde{\mu}_\theta(x_\lambda, \lambda)\right] \mathrm{d}\lambda.$$

where $\tilde{\mu}_\theta(x, \lambda) := \mu_\theta(x, t)$ and $\tilde{q}(x_\lambda|x_0) := q(x_t|x_0)$ for $t = \log\text{-SNR}^{-1}(\lambda)$. This shows that the only effect $\alpha_t$ has on the loss is through the values of the SNR at the endpoints. Still, because we draw uniform samples of $t$ to estimate the integral, the choice of masking schedule affects the variance.

**Multidimensional data.** In the previous sections, $x_t$ was assumed to be a single discrete token. To extend the method to multidimensional data, let $x_t$ be now a sequence $(x_t^{(1)}, x_t^{(2)}, \dots, x_t^{(N)})$, where each element $x_t^{(n)}$ represents a discrete token. We select a forward process which factorizes across all $N$ tokens: $q(x_t|x_s) = \prod_{n=1}^{N} q(x_t^{(n)}|x_s^{(n)})$. As a result, the forward marginals $q(x_t|x_0)$ and reversal $q(x_s|x_t, x_0)$ also factorize. In this case, we define the reverse model as $p_\theta(x_s|x_t) \triangleq \prod_{n=1}^{N} q(x_s^{(n)}|x_t^{(n)}, \mu_\theta^{(n)}(x_t, t))$, where $\mu_\theta(x_t, t)$ is a neural network that takes the full $N$ tokens as input and outputs $N$ probability vectors.[2] The $n$-th output $\mu_\theta^{(n)}(x_t, t)$ is a prediction model for $\mathbb{E}[x_0^{(n)}|x_t]$, the mean value of the $n$-th token. Repeating above derivations gives

$$\mathcal{L}_\infty^{(N)} \triangleq \int_0^1 \frac{\alpha_t'}{1 - \alpha_t}\mathbb{E}_{q(x_t|x_0)}\left[\sum_{n:x_t^{(n)}=m}(x_0^{(n)})^\top \log \mu_\theta^{(n)}(x_t, t)\right] \mathrm{d}t. \tag{5}$$

We term our simple masked diffusion model trained with loss (5) **MD4** (Masked Discrete Diffusion for Discrete Data). A single step of MD4 training algorithm is described in Alg. 1 in Appendix.

## 4  Sampling

We use ancestral sampling from our discrete-time reverse process for generation. We have found this yields slightly higher sample quality compared to other methods such as Euler discretization [29, 32]. For conditional generation tasks such as infilling, we find that the simple approach works best — we keep the conditioning tokens unmasked throughout the generation process. A complete description of the sampling algorithm can be found in Alg. 2 in Appendix.

---

[2]We intentionally choose the reverse model to factorize across dimensions because the true reverse transition $q(x_s|x_t)$ factorizes in the continuous-time limit (as $s$ approaches $t$).

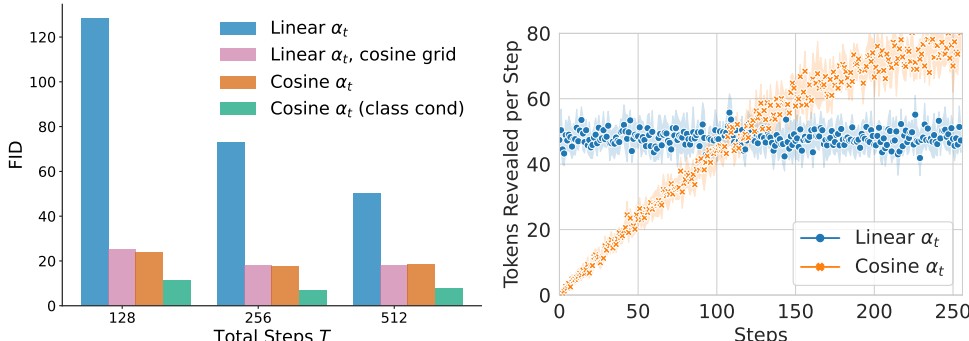

Figure 2: Left: FID evaluation for 50k samples randomly generated from MD4 on pixel-level modeling of ImageNet 64×64 (numbers in Tab. 6). Right: Number of tokens revealed per generation step ($T = 256$). Each image consists of $64 \times 64 \times 3 = 12288$ tokens.

**Impact of schedules and discretization.** For comparing different sampling configurations, we primarily use the FID score [40] on image datasets as our evaluation metric. We favor it over text generative perplexity[3] used in prior work [32], as the latter can be misleadingly reduced by lowering sample diversity [41]. We initially trained our model using the linear schedule, which achieves the best final ELBO overall; however, we found that sampling did not perform well with a standard uniform discretization grid $t(i) = \frac{i}{T}$. We hypothesize that time discretization can lead to conflicts by generating multiple tokens in a single step. We then switched to the cosine schedule (Tab. 4) that slows down unmasking at the beginning of reverse process. This drastically improves the FID on ImageNet 64×64 from 70 to 17 for $T = 256$ steps (Fig. 2, left). Building on this observation, we suggest using a "cosine" discretization grid for sampling in models trained with a linear schedule:

$$t(i) = \cos\left(\frac{\pi}{2}\left(1 - \frac{i}{T}\right)\right). \tag{6}$$

This induces the same discretization in $\alpha_t$ as the cosine schedule with a uniform grid, leading to comparable sample quality, as shown in Fig. 2 (left). In Fig. 2 (right), we plot the number of tokens unmasked per step for linear and cosine schedules with a uniform grid. We believe the cosine schedule performs better because it leverages information redundancy: with more tokens revealed, the remaining tokens become more predictable, reducing conflicts when unmasking them in a single step.

Although these findings were originally developed on images, we find them translate well to text (see Fig. 10). we expect other techniques such as top-$p$ sampling [41], classifier-free guidance [42, 43], and predictor-correctors [29, 44] to further improve sample quality of our models. While we reserve these for future work, we note that the JAX [45] implementation of categorical sampling implicitly truncates small probabilities, creating a similar effect to top-$p$ sampling. See App. G for details.

## 5 Relation to Existing Work

We discuss how to unify several existing masked diffusion models using our framework.

**Continuous-Time Markov Chains (CTMC).** To show the connection with the CTMC view presented in Austin et al. [14], Campbell et al. [29], we can write out the forward and reverse masked diffusion using CTMC machinery. To see this, for a short time $\Delta t$, given $x_0$, the Taylor expansions of our forward and reverse transition matrices at $t$ are

$$\bar{Q}(t, t + \Delta t) = I + Q(t)\Delta t + o(\Delta t) \quad \text{for} \quad Q(t) \triangleq \beta(t)(\mathbf{1}e_m^\top - I), \tag{7}$$

$$\bar{R}^{x_0}(t, t - \Delta t) = I + R^{x_0}(t)\Delta t + o(\Delta t) \quad \text{for} \quad R^{x_0}(t) \triangleq -\frac{\alpha_t'}{1 - \alpha_t}e_m(x_0 - e_m)^\top, \tag{8}$$

where $Q(t)$ and $R^{x_0}(t)$ are known as the *transition rate* matrices. Austin et al. [14] derived the same $Q(t)$ in App. A.6 of their paper. However, they did not explore the reverse process or a

---

[3]Perplexity of generated samples scored by a large language model such as GPT-2.

continuous-time objective. Campbell et al. [29] derived an alternative ELBO expression using rate matrices, which Kitouni et al. [46] further simplified for absorbing diffusion. In App. H.1, we show how to recover their expression by separating out a constant from our ELBO expression (4) and applying a discrete "integration-by-part". A key limitation of their expression is that it needs $N$ evaluations of the prediction model $\mu_\theta(\cdot, t)$ to compute an inner summation. To circumvent this computational burden, they used a doubly stochastic estimate. However, this leads to significantly higher variance compared to the analytic cross-entropy (4) which only requires one pass of $\mu_\theta(\cdot, t)$. Please refer to App. H.2 for more details.

**Score parameterization.** While so far we used a prediction model $\mu_\theta(x_t, t)$ for the mean of clean data given $x_t$ (i.e., mean parameterization), one can choose other ways of parameterizing the reverse model. Lou et al. [32], Benton et al. [35] proposed to parameterize the discrete "score" $s(x_t, t)_j \triangleq \frac{q_t(j)}{q_t(x_t)}$ and introduced a score-based loss for discrete diffusions. In App. H.3, we provide an alternative derivation of their loss which is simpler. We show the link between score and mean parameterizations through the following proposition.

**Proposition 1** (Score Parameterization vs. Mean Parameterization). *Let $q_t$ be the marginal distribution of the masked diffusion defined in Sec. 2 at time $t$. The discrete score $s(x_t, t)_j = \frac{q_t(j)}{q_t(x_t)}$ for a mask state $x_t = m$ and $j \neq m$ can be expressed as*

$$s(m, t)_j = \frac{\alpha_t}{1 - \alpha_t} \mathbb{E}[x_0|x_t = m]^\top e_j, \text{ which satisfies } \sum_{j \neq m} s(m, t)_j = \frac{\alpha_t}{1 - \alpha_t}. \tag{9}$$

Proposition 1 (proved in App. H.3) implies that a reasonable score model for a mask state is

$$s_\theta(m, t)_j = \frac{\alpha_t}{1 - \alpha_t} \mu_\theta(m, t)_j. \tag{10}$$

Indeed, substituting (10) into the score-based loss of Lou et al. [32], Benton et al. [35] recovers our objective (4). In Lou et al. [32], the score is parameterized as a neural network without enforcing the constraint in (9). This means the learned reverse model can be incompatible with the forward process. We find that our parameterization, which enforces the constraint, leads to more stable training and better results.

**Any-order autoregressive models.** The continuous-time reverse process of our masked diffusion model can be viewed as an any-order autoregressive model (AO-ARM) [47]. To see this, we reorder the tokens according to the timing of their unmasking events in the reverse process. For all tokens, the cumulative distribution functions (CDFs) of unmasking times $\{\tau_n\}_{n=1}^N$ are identical and satisfy $P(\tau_n \leq t) = P(x_t^{(n)} = m) = 1 - \alpha_t$. As a result, the ordering is uniformly random across all possible arrangements, and the token prediction during each unmasking event represents a prediction step in AO-ARMs. This connection was initially pointed out in Hoogeboom et al. [48, App. C]. The relation between our simplified ELBO (5) and the AO-ARM objective is independently clarified by Ou et al. [36]. Despite this equivalence, our work demonstrates that the masking schedule $\alpha_t$ introduces a new degree of freedom in the design of such models. Variations in $\alpha_t$ can lead to different distributions of unmasking times, significantly impacting performance in diffusion-style parallel sampling under time discretization, as shown in Fig. 2.

**Other related work.** Due to space constraint, we defer the discussion on other related work, including MaskGIT [39], discrete flow matching [49], SDDM [30], Blackout diffusion [50] and SUNDAE [51], to App. H.4.

## 6 Generalization to State-dependent Masking Schedules

Consider a scenario where some tokens hold more significance than others and we would like to unmask them earlier in the process. To achieve this, we introduce state-dependent masking schedules, where the probability of unmasking a token depends not only on time, but also on the token's value.

We first define the forward process for a single token $x_t$. Let $\alpha_t$ be a $m + 1$ dimensional vector function, i.e., there is a different function $\alpha_{t,i}$ for each possible value $i$ of the token $x_t$. Also, by

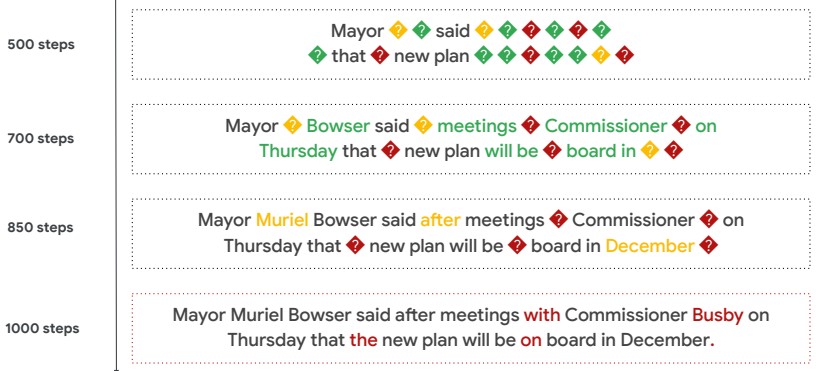

Figure 3: Iterative unmasking process for an unconditionally generated sample by MD4. This visualization only includes a subsequence from a generated sequence of 1024 tokens. "?" represents masks. Masked tokens are revealed sequentially: green (steps 500-700), yellow (700-850), and red (850-1000). Additional unconditional generation from MD4 can be found in App. K.5.

vector $\frac{\alpha_t}{\alpha_s}$ we denote the element-wise division of the two vectors. We define the forward transition as $q(x_t|x_s) = \mathrm{Cat}(x_t; \bar{Q}(s,t)^\top x_s)$ where

$$\bar{Q}(s,t) = \mathrm{diag}\Big(\frac{\alpha_t}{\alpha_s}\Big) + \Big(I - \mathrm{diag}\Big(\frac{\alpha_t}{\alpha_s}\Big)\Big)\mathbf{1}e_m^\top$$

and $\mathrm{diag}\big(\frac{\alpha_t}{\alpha_s}\big)$ is a diagonal matrix with the vector $\frac{\alpha_t}{\alpha_s}$ in its diagonal. The probability of moving from current state $x_s$ to a future state $x_t$ (either the same as $x_s$ or mask) is determined by a state-dependent rate $\big(\frac{\alpha_t}{\alpha_s}\big)^\top x_s$, while the marginal at time $s$ given $x_0$ is

$$q(x_s|x_0) = \mathrm{Cat}(x_s; \bar{Q}(s)^\top x_0) \quad \text{for} \quad \bar{Q}(s) = \mathrm{diag}(\alpha_s) + (I - \mathrm{diag}(\alpha_s))\mathbf{1}e_m^\top.$$

Further, for any time $0 \le s < t \le 1$ it holds that $q(x_t|x_0) = \sum_{x_s} q(x_t|x_s)q(x_s|x_0)$ so the above is a valid continuous-time Markov chain.

Given the forward conditionals and marginals, we can now compute the time reversal conditioned on $x_0$. The full form of $q(x_s|x_t, x_0)$ is derived in App. I.1. For $x_t = m$, we have

$$q(x_s|x_t = m, x_0) = q(x_s|x_t = m, x_0, x_0 x_0^\top) = \Big(\frac{\mathbf{1}-\alpha_s}{\mathbf{1}-\alpha_t}\Big)^\top x_0 e_m^\top x_s + \Big(\frac{\alpha_s - \alpha_t}{\mathbf{1}-\alpha_t}\Big)^\top x_0 x_0^\top x_s. \quad (11)$$

This suggests that the reverse model given $x_t = m$ can be chosen as $p_\theta(x_s|x_t = m) \triangleq q(x_s|x_t = m, \mu_\theta(x_t, t), \mathrm{diag}(\mu_\theta(x_t, t)))$ where $\mu_\theta(x_t, t)$ is a neural network that approximates $\mathbb{E}[x_0|x_t]$ while $\mathrm{diag}(\mu_\theta(x_t, t))$ approximates $\mathbb{E}[x_0 x_0^\top|x_t] = \mathrm{diag}(\mathbb{E}[x_0|x_t])$. We show in App. I.1 that the negative continuous-time ELBO for the state-dependent rate case is

$$\mathcal{L}_\infty = \int_0^1 \Big(\frac{\alpha_t'}{\mathbf{1}-\alpha_t}\Big)^\top \mathbb{E}_{q(x_t|x_0)}\big[\delta_{x_t,m} \cdot (x_0 - \mu_\theta(x_t, t) + x_0 x_0^\top \log \mu_\theta(x_t, t))\big] \mathrm{d}t. \quad (12)$$

Here, $\alpha_t'$ is the elementwise derivative of $\alpha_t$. This generalizes the MD4 loss (4), which is recovered when $\alpha_t$ is a scalar schedule times a vector of ones. For $N$ tokens, the model further generalize similarly to Sec. 3 and the loss is given in (32). We call this generalized model **GenMD4**.

To learn the token dependent masking schedule using ELBO optimization, we parametrize the $m+1$ dimensional function $\alpha_t$ using the polynomial schedule (see Fig. 1) as $\alpha_{t,i} = 1 - t^{w_i}$ and optimize each parameter $w_i > 0$.[4] The value of $w_i$, through the masking probability $1 - \alpha_{t,i}$, determines how fast the token with value $i$ jumps to the mask state. Since in the loss (12) the distribution $q(x_t|x_0)$ depends on $\alpha_t$ and thus the vector $w$, optimizing $w$ poses a discrete gradient estimation problem [see, e.g., 52]. Naive autodiff leads to biased gradients and pushes $w$ towards zero because the gradients cannot propagate through the (discrete) samples drawn from $q(x_t|x_0)$. To fix this, we used the REINFORCE leave-one-out estimator [53, 54] to compute low-variance unbiased gradients for optimizing $w$. Details are given in App. I.2.

---

[4]We only need $m$ learnable parameters $w_i$, for $i = 0, \ldots, m-1$, since $x_0$ can never be the mask token. For the final mask dimension we can choose an arbitrary fixed value such as $w_m = 0$.

Table 1: Zero-shot unconditional perplexity on five benchmark datasets from Radford et al. [57]. The numbers for other methods are from Lou et al. [32] except our reimplementation of SEDD Absorb. Our MD4 model achieves the best result on all benchmarks except LAMBADA where it is the second best. *The GPT-2 numbers are reported for the GPT-2 checkpoint pretrained on WebText instead of OWT thus is not a direct comparison.

| Size | Method | LAMBADA | WikiText2 | PTB | WikiText103 | IBW |
|---|---|---|---|---|---|---|
| Small | GPT-2 (WebText)* | **45.04** | 42.43 | 138.43 | 41.60 | 75.20 |
| | D3PM | $\leq 93.47$ | $\leq 77.28$ | $\leq 200.82$ | $\leq 75.16$ | $\leq 138.92$ |
| | Plaid | $\leq 57.28$ | $\leq 51.80$ | $\leq 142.60$ | $\leq 50.86$ | $\leq 91.12$ |
| | SEDD Absorb | $\leq 50.92$ | $\leq 41.84$ | $\leq 114.24$ | $\leq 40.62$ | $\leq 79.29$ |
| | SEDD Absorb (reimpl.) | $\leq 49.73$ | $\leq 38.94$ | $\leq 107.54$ | $\leq 39.15$ | $\leq 72.96$ |
| | MD4 (Ours) | $\leq 48.43$ | $\leq \mathbf{34.94}$ | $\leq \mathbf{102.26}$ | $\leq \mathbf{35.90}$ | $\leq \mathbf{68.10}$ |
| Medium | GPT-2 (WebText)* | **35.66** | 31.80 | 123.14 | 31.39 | 55.72 |
| | SEDD Absorb | $\leq 42.77$ | $\leq 31.04$ | $\leq 87.12$ | $\leq 29.98$ | $\leq 61.19$ |
| | MD4 (Ours) | $\leq 44.12$ | $\leq \mathbf{25.84}$ | $\leq \mathbf{66.07}$ | $\leq \mathbf{25.84}$ | $\leq \mathbf{51.45}$ |

# 7 Experiments

## 7.1 Text

Text is natural discrete data with rich structures. For comparison with prior work, we evaluate likelihood on two datasets: **text8** [55], a character-level text modeling benchmark, and **OpenWebText** [56], an open clone of the unreleased WebText dataset used to train GPT-2 [57]. We also assess our model's performance on downstream tasks by training on **FineWeb-Edu** [58], a high-quality dataset of fine educational text commonly used by the open-source community for comparing LLMs. Unless otherwise specified, a linear schedule and a cosine sampling grid are employed.

**OpenWebText.** We train MD4 of GPT-2 small (S) and GPT-2 medium (M) sizes on OpenWeb-Text and evaluate zero-shot perplexity on five benchmark datasets used in Radford et al. [57]. We keep our evaluation setup the same as SEDD [32]. To ensure fair comparison, we reimplemented SEDD in our codebase. Our implementation led to slightly better results than those reported in their paper.

As seen in Tab. 1, our small model outperforms previous best discrete diffusion models on all five tasks. We are also better than GPT-2 on all tasks except LAMBADA where we are the second best method. When scaling up to medium size, MD4 similarly beats SEDD and GPT-2 on 4 out of 5 tasks.

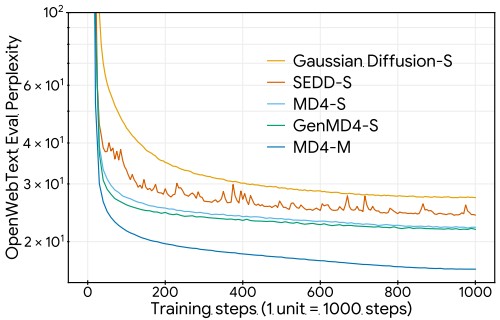

Figure 4: Perplexity on OpenWebText (OWT) validation set during training. The final numbers are reported in Tab. 5 in Appendix.

To confirm that the strong zero-shot performance stems from improved training, we plot perplexity on 2% OpenWebText validation set in Fig. 4. Our models converge faster and have better final likelihoods than prior methods. We also observed that SEDD [32] has training instabilities, likely due to score parameterization breaking consistency between forward and reverse processes (Sec. 5). Although GenMD4 achieves lower perplexity than MD4, we observed that the learned $w$s can overfit to dataset statistics, making it less effective on zero-shot transfer tasks.

We also assess our models' generation quality. Fig. 3 shows a randomly selected, notably coherent sample from MD4-medium and its denoising process. Fig. 10 demonstrates MD4's text infilling ability and highlights a substantial quality gain when transitioning from uniform to cosine discretization (see Sec. 4). Despite MD4's strong performance on quantitative metrics like generative perplexity, we have placed these results in Appendix Fig. 8 due to the metric's inherent unreliability, as noted in Sec. 4. We emphasize the more reliable FID-based assessments found in our image experiments.

Table 2: Bits Per Character (BPC) on Text8 test set. All models use standard 12-layer transformers similar to GPT-2 small [57] except Discrete Flow which uses $8 \times 3$ layers.

| Method | BPC ($\downarrow$) |
|---|---|
| *Continuous Diffusion* | |
| Plaid [22] (Our impl.) | $\leq 1.48$ |
| BFN [26] | $\leq 1.41$ |
| *Any-order Autoregressive* | |
| ARDM [48] | $\leq 1.43$ |
| MAC [61] | $\leq 1.40$ |
| *Autoregressive* | |
| IAF/SCF [62] | 1.88 |
| AR Argmax Flow [15] | 1.39 |
| Discrete Flow [59] | **1.23** |
| Transformer AR [14] | **1.23** |
| *Discrete Diffusion* | |
| Mult. Diffusion [15] | $\leq 1.72$ |
| D3PM Uniform [14] | $\leq 1.61$ |
| D3PM Absorb [14] | $\leq 1.45$ |
| SEDD Absorb [32] | $\leq 1.39$ |
| MD4 (Ours) | $\leq$ **1.37** |
| GenMD4 (Ours) | $\leq$ **1.34** |

Table 3: Bits Per Dimension (BPD) on CIFAR-10 test set and Downsampled ImageNet 64×64 [63] validation set. All models in the table are trained without data augmentation.

| | Method | #Params | BPD ($\downarrow$) |
|---|---|---|---|
| **CIFAR-10** | *Autoregressive* | | |
| | PixelRNN [63] | | 3.00 |
| | Gated PixelCNN [64] | | 3.03 |
| | PixelCNN++ [65] | 53M | 2.92 |
| | PixelSNAIL [66] | 46M | 2.85 |
| | Image Transformer [67] | | 2.90 |
| | Sparse Transformer [68] | 59M | 2.80 |
| | *Discrete Diffusion* | | |
| | D3PM Absorb [14] | 37M | $\leq 4.40$ |
| | D3PM Gauss [14] | 36M | $\leq 3.44$ |
| | Campbell et al. [29] | 36M | $\leq 3.59$ |
| | Campbell et al. [29] Absorb | 28M | $\leq 3.52$ |
| | MD4 (Ours) | 28M | $\leq$ **2.75** |
| **ImageNet 64×64** | *Autoregressive* | | |
| | PixelRNN [63] | | 3.63 |
| | Gated PixelCNN [64] | | 3.57 |
| | Sparse Transformer [68] | 152M | 3.44 |
| | Routing Transformer [69] | | 3.43 |
| | Perceiver AR [68] | 770M | **3.40** |
| | *Discrete Diffusion* | | |
| | MD4 (Ours) | 198M | $\leq$ **3.40** |

**Text8.** Following prior work [14, 32], we trained masked diffusion models on text8 and evaluate the bits-per-character on the test set (details in App. J.1). As seen in Tab. 2, our models outperform previous discrete and continuous diffusion models, as well as state-of-the-art AO-ARMs which are closely related to discrete diffusion [48]. Our model is only beaten by an autoregressive (AR) transformer and the AR-backbone Discrete Flow [59]. We believe this is because AR models only require learning a fixed generation order thus better utilize model capacity. Text8's small vocabulary (26 letters and a space) led us to expect limited flexibility from our state-dependent formulation. However, using the generalized objective in (12), GenMD4 achieved significantly better BPC than MD4, demonstrating the potential of state-dependent diffusion for discrete data.

**FineWeb-Edu.** We train MD4 on FineWeb-Edu and evaluate its zero-shot accuracy on the Hellaswag dataset [60], a popular common sense inference benchmark for LLMs. We directly compared MD4 to its AR counterparts – transformers with identical configurations (except for causal masking) trained on the same data. Results are summarized in Fig. 5.

MD4 demonstrates steady performance growth with increasing scale. While outperformed by AR models of the same size, the performance gap does not widen as model size increases. For example, AR-small reaches 30% accuracy in 50k steps, while MD4-small takes 200k steps (4x data efficiency difference). At the medium scale, AR achieves 37% in 270k steps, compared to MD4's 1 million steps.

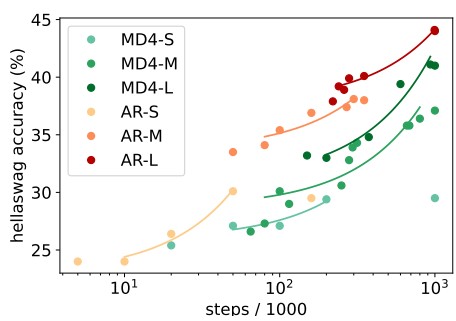

Figure 5: Hellaswag accuracy vs. training steps for MD4 and AR models at GPT-2 small, medium, and large scales.

## 7.2 Pixel-level image modeling

Unlike continuous diffusion which struggles with discrete data, we show that MD4, a discrete diffusion model, performs well on inherently continuous data, suggesting its potential for unifying

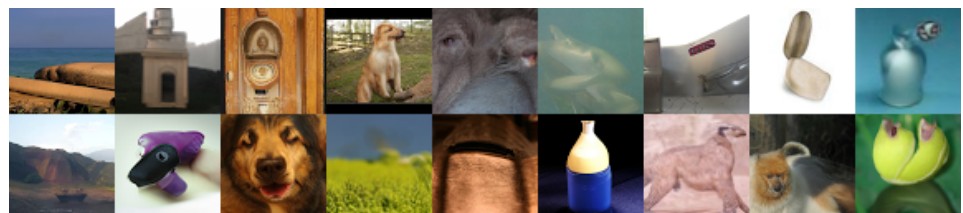

Figure 6: Non cherry-picked unconditional samples from MD4 trained on ImageNet 64x64, treating pixels as discrete tokens. More samples can be found in Fig. 9 in Appendix. The model is optimized for likelihood instead of visual quality—see e.g., Kingma et al. [33] for samples from a continuous diffusion model optimized similarly for likelihood.

modalities. We follow Austin et al. [14] and train MD4 on order-agnostic image data from CIFAR-10 and downsampled ImageNet $64\times64$ [63]. Each image is treated as a set of 256-valued discrete tokens, making the model agnostic to pixel proximity. We compare to other discrete diffusion and AR models with reported likelihood results on these datasets, although to our knowledge there are no published result on discrete diffusion for ImageNet $64 \times 64$ that directly model raw pixel space.

Tab. 3 summarizes our results. We establish a new state-of-the-art for discrete diffusion models, outperforming previous work [14, 29] by a significant margin. Our CIFAR-10 result surpasses the best reported AR result. On ImageNet $64 \times 64$, our results are competitive with Transformer AR models that are $4\times$ larger, as well as a strong continuous diffusion model VDM [33]. Notably, despite lacking knowledge of the ordinal structure of pixel values, MD4 outperforms models trained with this inductive bias, including D3PM Gauss and Campbell et al. [29] where the noising distribution is a discrete Gaussian that assigns larger probabilities to near pixel values. To isolate the differences caused by training objectives, we also implemented the Campbell et al. [29] objective with the absorbing process, showing its high variance hinders learning even with our architecture.

We provide a random sample from our ImageNet $64\times64$ model in Fig. 6. More results can be found in App. K. In Fig. 2, we plot the FID values of samples generated under different choices of schedules and discretization grids. We can see that the model with the linear schedule plus a cosine grid achieves an FID close to the model with cosine schedule, both significantly outperform the linear schedule with a uniform grid. We further trained a class-conditional model on ImageNet $64\times64$ that boosts the FID to around 7. Although these are not state-of-the-art FIDs on ImageNet $64\times64$, we emphasize our models are optimized for likelihood instead of sample quality.

## 8   Conclusion

In this work, we revisit masked diffusion models, focusing on a flexible continuous-time formulation. Existing works in this area are not easily accessible to non-specialists and present ELBOs that are difficult to optimize, often resulting in performance that is not competitive with continuous diffusions and AR models. The framework we propose provides a very simple expression of the ELBO as a weighted integral of cross-entropy losses. Additionally, we propose a generalized masked diffusion formulation (GenMD4), where the masking schedule depends on the current state of the process, and derive its corresponding ELBO. On text data, our MD4 models outperform existing discrete and continuous diffusion models. For pixel-level image modeling, we significantly improve discrete diffusion results, outperforming similar-sized AR models and achieving comparable likelihoods to continuous diffusion models such as VDM. GenMD4 provides further improvements in terms of likelihoods over the state-independent case.

Although we have improved masked diffusion models, they still suffer from limitations. First, in some tasks such as text8, masked diffusions are not yet competitive with AR models. We conjecture that this is because AR models can better leverage model capacity since they only require learning one order. It would be interesting to develop better architectures for discrete diffusions. Moreover, GenMD4 is promising, but it can easily overfit to the dataset, making it less effective for zero-shot transfer compared to simpler versions. Additionally, inference with a state-dependent schedule is more challenging.

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

Table 4: Masking schedule formulas.

| Masking schedules | $\alpha_t$ | Cross-entropy loss weight $\frac{\alpha'_t}{1-\alpha_t}$ |
|---|---|---|
| Linear | $1 - t$ | $-\frac{1}{t}$ |
| Polynomial | $1 - t^w$ | $-\frac{w}{t}$ |
| Geometric | $\exp\left(-\bar{\beta}_{\min}^{1-t}\bar{\beta}_{\max}^t\right)$ | $-\frac{\exp\left(-\bar{\beta}_{\min}^{1-t}\bar{\beta}_{\max}^t\right)}{1-\exp\left(-\bar{\beta}_{\min}^{1-t}\bar{\beta}_{\max}^t\right)}\bar{\beta}_{\min}^{1-t}\bar{\beta}_{\max}^t \log \frac{\sigma_{\min}}{\sigma_{\max}}$ |
| Cosine | $1 - \cos\left(\frac{\pi}{2}(1 - t)\right)$ | $-\frac{\pi}{2}\tan\left(\frac{\pi}{2}(1 - t)\right)$ |

## A  Discrete-time derivation

We divide time from 0 to 1 into $T$ intervals, and let $s(i) = (i - 1)/T$, $t(i) = i/T$. The forward transition matrix $Q_i \in \mathbb{R}^{(m+1)\times(m+1)}$ ($m$ is vocabulary size) at time $t(i)$ is

$$[Q_i]_{jk} = \begin{cases} 1 & j = k = m \\ 1 - \beta_i & j = k \neq m \\ \beta_i & k = m, j \neq m \\ 0 & \text{otherwise} \end{cases}$$

or more compactly written as

$$Q_i = (1 - \beta_i)I + \beta_i \mathbf{1}e_m^\top,$$

where $\mathbf{1}$ denotes an all-one vector of size $m + 1$, and $e_m$ is an one-hot vector of size $m + 1$ with the $m$-th element (recall that counting starts from 0) being one. We use an one-hot vector $x_t$ of length $m + 1$ to denote the discrete state. The forward conditionals are defined as

$$q(x_{t(i)}|x_{s(i)}) = \mathrm{Cat}(x_{t(i)}; Q_i^\top x_{s(i)}) = x_{s(i)}^\top Q_i x_{t(i)}, \tag{13}$$

where $Q_i^\top x_{s(i)}$ is the probabilities for each of the $m + 1$ categories that $x_{t(i)}$ can take. The marginal forward distribution at time $t(i)$ given $x_0$ is

$$q(x_{t(i)}|x_0) = \mathrm{Cat}(x_{t(i)}; \bar{Q}_i^\top x_0) = x_0^\top \bar{Q}_i x_{t(i)},$$

where $\bar{Q}_i = \prod_{j=1}^i Q_j = \prod_{j=1}^i (1 - \beta_j)I + \left(1 - \prod_{j=1}^i(1 - \beta_j)\right)\mathbf{1}e_m^\top$. To see what this leads to in continuous time, we let $\beta_i = \frac{\beta(t(i))}{T}$ and $T \to \infty$:

$$\prod_{j=1}^i(1 - \beta_j) = \exp\left(\sum_{j=1}^i \log(1 - \beta_j)\right)$$

$$= \exp\left(\sum_{j=1}^i -\frac{\beta(t(j))}{T} + o(1/T)\right)$$

$$\overset{T\to\infty}{\to} \exp\left(-\int_0^{t(i)} \beta(s)\mathrm{d}s\right).$$

We let $\bar{Q}(t)$ denote the limit of $\bar{Q}_i$ in this case:

$$\bar{Q}(t) = \exp\left(-\int_0^t \beta(s)\mathrm{d}s\right)I + \left(1 - \exp\left(-\int_0^t \beta(s)\mathrm{d}s\right)\right)\mathbf{1}e_m^\top$$

$$\triangleq \alpha_t I + (1 - \alpha_t)\mathbf{1}e_m^\top.$$

Here we define $\alpha_t \triangleq \exp(-\int_0^t \beta(s)\mathrm{d}s)$. And the marginal forward transition is

$$q(x_t|x_0) = \mathrm{Cat}(x_t; \bar{Q}(t)^\top x_0) = x_0^\top \bar{Q}(t)x_t = \alpha_t x_0^\top x_t + (1 - \alpha_t)e_m^\top x_t. \tag{14}$$

# B Continuous-time derivation

We consider a continuous-time Markov chain with transition rates

$$Q(t) = (Q_i - I)/(1/T) = \beta(t)(\mathbf{1}e_m^\top - I). \tag{15}$$

For simplicity, we let $Q = \mathbf{1}e_m^\top - I$. The marginal forward distribution at time $t$ given $x_0$ is $q(x_t|x_0) = \text{Cat}(x_t; \bar{Q}(t)^\top x_0)$, where

$$\bar{Q}(t) = \exp\left(\int_0^t Q(s)\mathrm{d}s\right) = \exp\left(Q\int_0^t \beta(s)\mathrm{d}s\right) = \exp(\bar{\beta}(t)Q).$$

Here we define $\bar{\beta}(t) \triangleq \int_0^t \beta(s)\mathrm{d}s$. The matrix exponential can be computed via eigendecomposition:

$$\bar{\beta}(t)Q = U\Lambda U^{-1},$$

where

$$U = I - e_m e_m^\top + \frac{1}{\sqrt{n+1}}\mathbf{1}e_m^\top,$$
$$U^{-1} = I + \sqrt{n+1}e_m e_m^\top - \mathbf{1}e_m^\top,$$
$$\Lambda = \bar{\beta}(t)(e_m e_m^\top - I),$$

and thus $\exp(\Lambda) = \alpha_t I + (1 - \alpha_t)e_m e_m^\top$,

$$\bar{Q}(t) = U\exp(\Lambda)U^{-1} = \alpha_t I + (1 - \alpha_t)\mathbf{1}e_m^\top.$$

A simpler derivation uses the following property:

$$Q^2 = -Q.$$

Therefore,

$$\begin{aligned}
\bar{Q}(t) &= \exp(\bar{\beta}(t)Q) \\
&= I + \bar{\beta}(t)Q + \frac{1}{2}\bar{\beta}(t)^2 Q^2 + \frac{1}{3}\bar{\beta}(t)^3 Q^3 + \dots \\
&= I + Q - (1 - \bar{\beta}(t) + \frac{1}{2}\bar{\beta}(t)^2 - \frac{1}{3}\bar{\beta}(t)^3 + \dots)Q \\
&= I + Q - \exp(-\bar{\beta}(t))Q \\
&= \alpha_t I + (1 - \alpha_t)\mathbf{1}e_m^\top.
\end{aligned}$$

This marginal forward transition matrix at time $t$ coincides with the result (1) we get by taking the limit of discrete-time derivation.

**Arbitrary discretization of the continuous-time forward process.** For the discrete-time process we have defined the per-step transition in (13). For the continuous-time process, we can derive the transition matrix $\bar{Q}(s,t)_{ij} \triangleq q(x_t = j|x_s = i)$ between two arbitrary time $s$ and $t$ as the solution to the following differential equation (known as Kolmogorov forward equation)

$$\frac{\mathrm{d}}{\mathrm{d}t}\bar{Q}(s,t) = \bar{Q}(s,t)Q(t) \text{ where } Q(t) = \beta(t)Q$$

with initial condition $\bar{Q}(s,s) = I$. The solution is given by

$$\bar{Q}(s,t) = \exp\left((\bar{\beta}(t) - \bar{\beta}(s))Q\right) = \bar{Q}(s)^{-1}\bar{Q}(t).$$

Routine work (using the Woodbury matrix inversion lemma) shows that

$$\bar{Q}(t)^{-1} = \alpha_t^{-1}I + (1 - \alpha_t^{-1})\mathbf{1}e_m^\top.$$

Plugging the result back, we get the forward transition distribution from $s$ to $t$:

$$q(x_t|x_s) = \text{Cat}(x_t; \bar{Q}(s,t)^\top x_s) = x_s^\top \bar{Q}(s,t)x_t, \tag{16}$$
$$\text{where } \bar{Q}(s,t) \triangleq \bar{Q}(s)^{-1}\bar{Q}(t) = \frac{\alpha_t}{\alpha_s}I + \left(1 - \frac{\alpha_t}{\alpha_s}\right)\mathbf{1}e_m^\top.$$

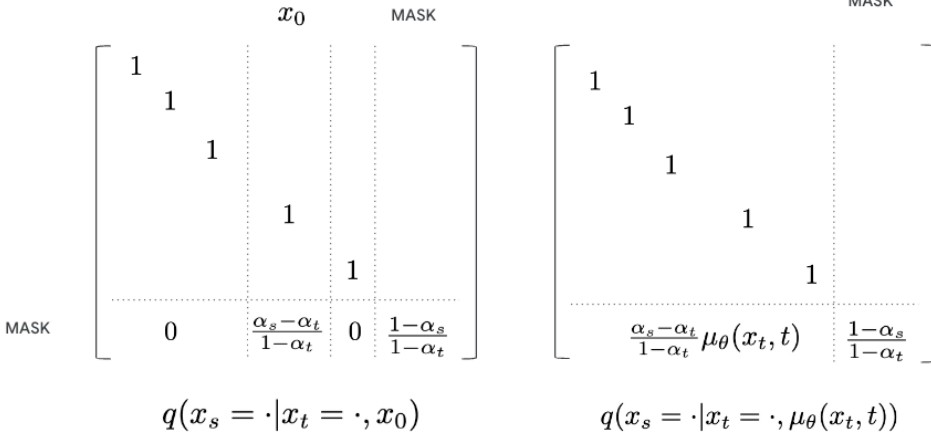

$$q(x_s = \cdot | x_t = \cdot, x_0) \qquad\qquad q(x_s = \cdot | x_t = \cdot, \mu_\theta(x_t, t))$$

Figure 7: The reverse transition probability and our generative model. Left: $q(x_s = \cdot | x_t = \cdot, x_0)$ in matrix form where first index is $x_t$ and second index is $x_s$. Right: $p_\theta(x_s = \cdot | x_t = \cdot) \triangleq q(x_s = \cdot | x_t = \cdot, \mu_\theta(x_t, t))$ also in matrix form.

## C   Time reversal of the forward process given $x_0$

The analytic property of our forward process allows to compute many quantities of interest in closed form. One such quantity frequently used in diffusion models is the time reversal of the forward process given $x_0$: $q(x_s | x_t, x_0)$. We can compute it using (14) and (16) as

$$q(x_s | x_t, x_0) = \frac{q(x_t | x_s) q(x_s | x_0)}{q(x_t | x_0)}$$

$$= \begin{cases} \frac{\alpha_s - \alpha_t}{1 - \alpha_t} x_s^\top x_0 & x_s \neq m, x_t = m \\ \frac{1 - \alpha_s}{1 - \alpha_t} & x_s = m, x_t = m \\ x_s^\top x_t & x_t \neq m. \end{cases} \qquad (17)$$

Visually, eqn (17) is a $\mathbb{R}^{(m+1) \times (m+1)}$ matrix (Fig. 7, left) whose first index is $x_t$ and the second is $x_s$. The matrix is almost an identity matrix except the last row corresponding to $x_t$ is the mask token. The last row means with probability of $\frac{\alpha_s - \alpha_t}{1 - \alpha_t}$ the mask token gets unmasked to become $x_0$, and with probability of $\frac{1 - \alpha_s}{1 - \alpha_t}$ it remains masked.

Alternatively, we can rewrite the above using reverse transition matrix $\bar{R}^{x_0}(t, s) \in \mathbb{R}^{(m+1) \times (m+1)}$ as

$$q(x_s | x_t, x_0) = \mathrm{Cat}(x_s; \bar{R}^{x_0}(t, s)^\top x_t), \text{ where } \bar{R}^{x_0}(t, s) = I + \frac{\alpha_s - \alpha_t}{1 - \alpha_t} e_m (x_0 - e_m)^\top.$$

We are also interested in what would happen in the infinitesimal time limit, i.e., when $s = t - \Delta t$ and $\Delta t \to 0$. Note that

$$\alpha_{t - \Delta t} - \alpha_t = -\alpha_t' \Delta t + o(\Delta t).$$

Plugging it into the original formula, we get

$$\bar{R}^{x_0}(t, t - \Delta t) = I - \frac{\alpha_t'}{1 - \alpha_t} e_m (x_0 - e_m)^\top \Delta t + o(\Delta t).$$

Comparing the above with the transition rate matrix $R^{x_0}(t)$ definition

$$\bar{R}^{x_0}(t, t - \Delta t) = I + R^{x_0}(t) \Delta t + o(\Delta t),$$

we have determined the transition rate matrix for the reverse process conditioned on $x_0$:

$$R^{x_0}(t) = -\frac{\alpha_t'}{1 - \alpha_t} e_m (x_0 - e_m)^\top. \qquad (18)$$

# D  Details of the ELBO

Using (17) and (3), we compute the KL divergences between forward and reverse transitions

$$\mathrm{KL}(q(x_s|x_t, x_0) \| p_\theta(x_s|x_t)) = \mathrm{KL}(q(x_s|x_t, x_0) \| q(x_s|x_t, \mu_\theta(x_t, t))) \tag{19}$$

$$= \begin{cases} \sum_{x_s=0}^m q(x_s|x_t, x_0) \log \frac{q(x_s|x_t, x_0)}{q(x_s|x_t, \mu_\theta(x_t, t))} & x_t = m \\ 0 & x_t \neq m \end{cases}$$

$$= \delta_{x_t=m} \sum_{k \neq m} \frac{\alpha_s - \alpha_t}{1 - \alpha_t} x_0^\top e_k \log \frac{x_0^\top e_k}{\mu_\theta(x_t, t)^\top e_k}$$

$$= -\delta_{x_t=m} \frac{\alpha_s - \alpha_t}{1 - \alpha_t} x_0^\top \log \mu_\theta(x_t, t).$$

Note that $0 \log 0 = 0$. Alternatively, this result can be easily obtained from the visual depictions of $q(x_s|x_t, x_0)$ and $p_\theta(x_s|x_t)$ shown in Fig. 7. In this case, the reconstruction term becomes

$$\mathbb{E}_{q(x_{t(1)}|x_0)}[\log p(x_0|x_{t(1)})] = \sum_{k=0}^m q_{t(1)|0}(k|x_0) \log \frac{q_{t(1)|0}(k|x_0)}{\sum_{j \neq m} q_{t(1)|0}(k|j)}$$

$$= \alpha_{t(1)} \cdot \log \frac{\alpha_{t(1)}}{\alpha_{t(1)}} + (1 - \alpha_{t(1)}) \log \frac{1}{m}$$

$$= -(1 - \alpha_{t(1)}) \log m.$$

The prior KL term can be computed as

$$\mathrm{KL}(q(x_1|x_0) \| p(x_1)) = \mathrm{KL}(\delta_{x_1, m} \| \delta_{x_1, m}) = 0.$$

As usual, we take the continuous-time limit by letting $T \to \infty$:

$$\mathcal{L}_\infty \triangleq \lim_{T \to \infty} \mathcal{L}_T$$

$$= \lim_{T \to \infty} \sum_{i=2}^T -\frac{\alpha_{s(i)} - \alpha_{t(i)}}{s(i) - t(i)} \frac{s(i) - t(i)}{1 - \alpha_{t(i)}} x_0^\top \mathbb{E}_{q(x_{t(i)}|x_0)} \left[ \delta_{x_{t(i)}, m} \log \mu_\theta(x_{t(i)}, t(i)) \right]$$

$$= \int_{t(1)}^1 \frac{\alpha_t'}{1 - \alpha_t} x_0^\top \mathbb{E}_{q(x_t|x_0)} \left[ \delta_{x_t, m} \log \mu_\theta(x_t, t) \right] dt.$$

# E  Avoiding undefined KL divergence

When defining the forward process, we often do not want $\alpha_1$ to be exactly 0, or equivalently, $\lambda_1$ to be $\infty$ for numerical stability reasons. Instead, we set $\lambda_1$ to be a finite value, and thereby $\alpha_1$ has a small positive value. This has a problem that the support of $q(x_1|x_0)$ is no longer $\{m\}$ and instead becomes $\{m, x_0\}$. As a result, the KL divergence between $q(x_1|x_0)$ and $p(x_1)$ is undefined because $q(x_1|x_0)$ is not absolutely continuous with respect to $p(x_1) = \delta_{x_1, m}$. To resolve the issue, we modify the prior distribution $p(x_1)$ such that it has support over all $m + 1$ values. One such choice is letting

$$p(x_1) = \frac{\alpha_1}{m} \sum_{j \neq m} \delta_{x_1, j} + (1 - \alpha_1) \delta_{x_1, m}.$$

Then, the prior KL divergence term becomes

$$\mathrm{KL}(q(x_1|x_0) \| p(x_1)) = \sum_{x_1=0}^m q(x_1|x_0) \log \frac{q(x_1|x_0)}{p(x_1)}$$

$$= \sum_{x_1=0}^m (\alpha_1 \delta_{x_1, x_0} + (1 - \alpha_1) \delta_{x_1, m}) \log \frac{\alpha_1 \delta_{x_1, x_0} + (1 - \alpha_1) \delta_{x_1=m}}{p(x_1)}$$

$$= \alpha_1 \log \frac{\alpha_1}{\alpha_1/m} + (1 - \alpha_1) \log \frac{1 - \alpha_1}{1 - \alpha_1}$$

$$= \alpha_1 \log m.$$

# F Details of Training and Sampling with MD4

## F.1 Training

---
**Algorithm 1** A single step of training with MD4.

---
**Input:** data minibatch $\{x_t^i\}_{i=1}^B$, network $\mu_\theta(\cdot, t)$, masking schedule $\alpha_t$
**for** $i = 1, \ldots, B$ **do** (in parallel):
    $t_i \leftarrow \mod(u + i/B, 1)$, $u \sim U[0, 1]$
    for $n \in [N]$, mask out each token $x_0^{i,(n)}$ independently with probability $1 - \alpha_{t_i}$ to obtain $x_{t_i}^i$
    for $n \in [N]$, if $x_{t_i}^{(n)} = m$, compute weighted cross entropy loss $\frac{\alpha'_{t_i}}{1 - \alpha_{t_i}} (x_0^{i,(n)})^\top \log \mu_\theta^{(n)}(x_{t_i}^i, t_i)$

Sum over all weighted cross entropy losses for mask positions and optimize via autodiff

---

## F.2 Sampling

---
**Algorithm 2** Unconditional and conditional generation (e.g., infilling) with MD4.

---
**Input:** Context sequence $x^c$ of length $N$, with masks indicating the target areas for generation
**Init:** $\{t(i)\}_{i=0}^T \leftarrow \text{discretize}([0, 1])$, $x_{t(T)} \leftarrow x^c$
**for** $i = T, T - 1, \ldots, 1$ **do**
    $t \leftarrow t(i)$, $s \leftarrow t(i - 1)$
    for $n \in [N]$, if $x_t^{(n)} = m$, draw $x_s^{(n)} \sim \text{Cat}(\frac{\alpha_s - \alpha_t}{1 - \alpha_t} \mu_\theta^{(n)}(x_t, t) + \frac{1 - \alpha_s}{1 - \alpha_t} e_m)$ else $x_s^{(n)} \leftarrow x_t^{(n)}$
**return** $x_0$.

---

# G JAX Categorical Sampling and Implicit Top-$p$

We noticed that the following equivalent implementation of Alg. 2 leads to significantly worse sample quality in JAX:

---
**Algorithm 3** Variant of Alg. 2 that yields lower sample quality when implemented in JAX.

---
**Input:** Token sequence $x^c$ of length $N$, with masks indicating the target areas for generation
**Init:** $\{t(i)\}_{i=0}^T \leftarrow \text{discretize}([0, 1])$, $x_{t(T)} \leftarrow x^c$
**for** $i = T, T - 1, \ldots, 1$ **do**
    $t \leftarrow t(i)$, $s \leftarrow t(i - 1)$
    **for** $n \in [N]$ **do** (in parallel)
        draw $u \sim U[0, 1]$
        **if** $x_t^{(n)} = m$ **and** $u < \frac{\alpha_s - \alpha_t}{1 - \alpha_t}$ **then**
            draw $x_s^{(n)} \sim \text{Cat}(\mu_\theta^{(n)}(x_t, t))$
        **else**
            $x_s^{(n)} \leftarrow x_t^{(n)}$
**return** $x_0$.

---

However, mathetically it is equivalent to Alg. 2 and should produce identical results. Our investigation revealed that the issue arises because Alg. 2 scales the output probabilities of $\mu_\theta$ by a small factor $\frac{\alpha_s - \alpha_t}{1 - \alpha_t}$ as $s$ is close to $t$, causing some categories to have very low probabilities. JAX, however, implements categorical sampling using Gumbel argmax, which is less numerically stable than methods like binary search. As a result, categories with low probabilities are rarely sampled, even when their cumulative probability is significant. In our experiment, we found that categories with probabilities below 1e-8 are rarely sampled out of a total of 50K categories. Thus, Alg. 2 implicitly performs top-$p$ sampling (with a dynamic p) under JAX's categorical sampling, yielding better sample quality than Alg. 3 where $\mu_\theta$ is not scaled by a small factor and has fewer categories truncated.

# H Unifying Existing Masked Diffusion Models

## H.1 The CTMC point of view

We first prove a lemma that connects the forward and reverse transition rate matrices. This follows from the results in [29] but we give a proof for completeness.

**Lemma 2.** *The forward transition rate matrix $Q(t)$ and the reverse transition rate matrix (given $x_0$) $R^{x_0}(t)$ satisfy:*

$$R^{x_0}(t)_{kj} = Q(t)_{jk} \frac{q_{t|0}(j|x_0)}{q_{t|0}(k|x_0)} \text{ for } j \neq k. \tag{20}$$

**Proof** Consider the reverse transition from time $t + \tau$ to $t$. For $j \neq k$, Bayes' rule yields

$$
\begin{aligned}
q(x_t = j | x_{t+\tau} = k, x_0) &= \frac{q(x_t = j | x_0) q(x_{t+\tau} = k | x_t = j)}{q(x_{t+\tau} = k | x_0)} \\
&= \frac{q(x_t = j | x_0)(\delta_{jk} + Q(t)_{jk}\tau + o(\tau))}{q(x_{t+\tau} = k | x_0)} \\
&\overset{\tau \to 0}{=} \delta_{kj} + \frac{q(x_t = j | x_0)}{q(x_t = k | x_0)} Q(t)_{jk}\tau + o(\tau).
\end{aligned}
$$

Then, it follows from the definition of the transition rate matrix that $R^{x_0}(t)_{kj} = Q(t)_{jk}\frac{q_{t|0}(j|x_0)}{q_{t|0}(k|x_0)}$. $\quad\square$

**Proposition 3.** *We use the shorthand $R_\theta(t)_{kj}$ to denote the approximate reverse transition rate from the state $k$ to $j$ obtained by substituting our prediction model $\mu_\theta(k)$ for $x_0$ in $R^{x_0}(t)_{kj}$. Then, the continuous-time objective* (4) *can be equivalently expressed as*

$$\mathcal{L}_\infty = -\int_{t(1)}^1 \mathbb{E}_{q_{t|0}(k|x_0)}\Big[R_\theta(t)_{kk} + \sum_{j \neq k} Q(t)_{kj} \log R_\theta(t)_{jk}\Big] \mathrm{d}t + C, \tag{21}$$

*where $C$ is a constant independent of $\theta$.*

**Proof** To rewrite our objective $\mathcal{L}_\infty$ with the transition rate matrices, we first go back to (19). There, instead of plugging in the explicit form of $\bar{R}^{x_0}(t,s)$, we substitute it with (8) which leverages the transition rate $R^{x_0}(t)$. To simplify the notation, we assume $x_t = k$ and use the shorthand $R_\theta(t)_{kj} \triangleq R^{\mu_\theta(k)}(t)_{kj}$. We then have

$$
\begin{aligned}
&\mathrm{KL}(q(x_{t-\Delta t}|x_t, x_0) \| p_\theta(x_{t-\Delta t}|x_t)) \\
&= \mathrm{KL}(\mathrm{Cat}(x_s; \bar{R}^{x_0}(t, t-\Delta t)^\top e_k) \| \mathrm{Cat}(x_s; \bar{R}^{\mu_\theta(k)}(t, t-\Delta t)^\top e_k)) \\
&= \sum_{j=0}^m e_k^\top (I + R^{x_0}(t)\Delta t + o(\Delta t)) e_j \log \frac{e_k^\top (I + R^{x_0}(t)\Delta t + o(\Delta t)) e_j}{e_k^\top (I + R_\theta(t)\Delta t + o(\Delta t)) e_j} \\
&= (1 + R^{x_0}(t)_{kk}\Delta t) \log \frac{1 + R^{x_0}(t)_{kk}\Delta t + o(\Delta t)}{1 + R_\theta(t)_{kk}\Delta t + o(\Delta t)} \\
&\quad + \sum_{j \neq k} (R^{x_0}(t)_{kj}\Delta t) \log \frac{R^{x_0}(t)_{kj}\Delta t + o(\Delta t)}{R_\theta(t)_{kj}\Delta t + o(\Delta t)} + o(\Delta t) \\
&= (R^{x_0}(t)_{kk} - R_\theta(t)_{kk})\Delta t + \sum_{j \neq k} (R^{x_0}(t)_{kj}\Delta t) \log \frac{R^{x_0}(t)_{kj}\Delta t + o(\Delta t)}{R_\theta(t)_{kj}\Delta t + o(\Delta t)} + o(\Delta t).
\end{aligned}
$$

For the last identity, we have used the fact that $\log(1+x) = x + o(x)$. To obtain $\mathcal{L}_\infty$, we take the limit of $\mathcal{L}_T$ as $T \to \infty$, which is equivalent to letting $\Delta t = 1/T \to 0$. We obtain

$$
\begin{aligned}
\mathcal{L}_\infty &= \lim_{T \to \infty} \sum_{i=2}^T \mathbb{E}_{q(x_{t(i)}|x_0)}[\mathrm{KL}(q(x_{s(i)}|x_{t(i)}, x_0) \| p_\theta(x_{s(i)}|x_{t(i)})))] \\
&= \lim_{T \to \infty} \sum_{i=2}^T \mathbb{E}_{q(x_{t(i)}|x_0)} \Big[ \Big( R^{x_0}(t(i))_{kk} - R_\theta(t(i))_{kk} \\
&\quad + \sum_{j \neq k} R^{x_0}(t(i))_{kj} \log \frac{R^{x_0}(t(i))_{kj} \Delta t + o(\Delta t)}{R_\theta(t(i))_{kj} \Delta t + o(\Delta t)} \Big) \Delta t + o(\Delta t) \Big] \\
&= \int_{t(1)}^1 \mathbb{E}_{q_{t|0}(k|x_0)} \Big[ R^{x_0}(t)_{kk} - R_\theta(t)_{kk} + \sum_{j \neq k} R^{x_0}(t)_{kj} \log \frac{R^{x_0}(t)_{kj}}{R_\theta(t)_{kj}} \Big] dt.
\end{aligned}
$$

Note that $R^{x_0}(t)$ is a constant matrix independent of $\theta$. Absorbing all constant terms into $C$, we have

$$
\mathcal{L}_\infty = - \int_{t(1)}^1 \mathbb{E}_{q_{t|0}(k|x_0)} \Big[ R_\theta(t)_{kk} + \sum_{j \neq k} R^{x_0}(t)_{kj} \log R_\theta(t)_{kj} \Big] dt + C.
$$

Next, we subtitute $R^{x_0}(t)$ with the forward transition rate using Lemma 2:

$$
\begin{aligned}
\mathcal{L}_\infty &= - \int_{t(1)}^1 \mathbb{E}_{q_{t|0}(k|x_0)} \Big[ R_\theta(t)_{kk} + \sum_{j \neq k} Q(t)_{jk} \frac{q_{t|0}(j|x_0)}{q_{t|0}(k|x_0)} \log R_\theta(t)_{kj} \Big] dt + C \\
&= - \int_{t(1)}^1 \Big[ \sum_{k=0}^m q_{t|0}(k|x_0) R_\theta(t)_{kk} + \sum_{k=0}^m \sum_{j \neq k} Q(t)_{jk} q_{t|0}(j|x_0) \log R_\theta(t)_{kj} \Big] dt + C \\
&= - \int_{t(1)}^1 \Big[ \sum_{k=0}^m q_{t|0}(k|x_0) R_\theta(t)_{kk} + \sum_{k=0}^m \sum_{j \neq k} Q(t)_{kj} q_{t|0}(k|x_0) \log R_\theta(t)_{jk} \Big] dt + C,
\end{aligned}
$$

where the last identity used the discrete analog to integration-by-part (or summation-by-part): $\sum_{k=0} \sum_{j \neq k} f(j, k) = \sum_{k=0} \sum_{j \neq k} f(k, j)$. Rearranging the terms then gives (21). $\qquad \square$

## H.2  Differences from Campbell et al. [29]

Campbell et al. [29] used the first term of (21) as the training loss. A key limitation of this loss function is from the inner summation term

$$
\sum_{j \neq k} Q(t)_{kj} \log R_\theta(t)_{jk}.
$$

For single dimension case, the sum is analytically computable due to the sparse structure of $R_\theta(t)$—if $x_t = k$ is mask, the second term disappears; otherwise the only possible neighbor $j$ is a mask. However, for multidimensional data, $j$ will represent all $N-1$ neighbors in the forward process, i.e., the states we get from mask out a single unmasked dimension of $x_t = k$. Recall that $R_\theta(t)_{jk}$ is computed as substituting our neural network prediction model $\mu_\theta(j)$ for $x_0$ in $R^{x_0}(t)_{jk}$. Therefore, the summation together with $R_\theta(t)_{kk}$ requires $N$ evaluations of $\mu_\theta(\cdot)$. This is prohibitive since the neural network model is usually expensive. To resolve this issue, Campbell et al. [29] proposed to rewrite the sum as

$$
\mathbb{E}_{j \sim \tilde{q}(\cdot|k)} [Z_k \log R_\theta(t)_{jk}] \quad \text{where} \quad \tilde{q}(j|k) = \frac{Q(t)_{kj}}{Z_k}, Z_k \triangleq \sum_{j' \neq k} Q(t)_{kj'}
$$

and estimate it through Monte Carlo. Taking into account the outer expectation under $q_{t|0}(k|x_0)$, the computation of the loss then becomes a doubly stochastic estimate (using $k \sim q_{t|0}(k|x_0)$ and $j \sim \tilde{q}(j|k)$) which suffers from large variance. In contrast, the form of our loss (4) only requires evaluating $\mu_\theta$ once for a single stochastic estimation of the expectation w.r.t. $q(x_t|x_0)$.

## H.3 Score parameterization

We provide a simpler derivation of the score-based loss [32, 35] below. We start from the form of the ELBO in (21) and rewrite it as

$$\mathcal{L}_\infty = \int_{t(1)}^1 \mathbb{E}_{q_{t|0}(k|x_0)} \Big[ \sum_{j \neq k} \Big( R^{\mu_\theta}(t)_{kj} - R^{x_0}(t)_{kj} + R^{x_0}(t)_{kj} \log \frac{R^{x_0}(t)_{kj}}{R^{\mu_\theta}(t)_{kj}} \Big) \Big] \mathrm{d}t. \qquad (22)$$

For the last identity we used the zero-row-sum property of transition rate matrix:

$$R^{x_0}(t)_{kk} = - \sum_{j \neq k} R^{x_0}(t)_{kj}.$$

If we plug (20) into (22) and reparameterize with a score model

$$s_\theta(x_t)_j \triangleq \frac{q_{t|0}(j|\mu_\theta(x_t))}{q(x_t|\mu_\theta(x_t))}, \qquad (23)$$

we recover the score entropy loss function from Lou et al. [32], Benton et al. [35]:

$$\mathcal{L}_\infty = \int_{t(1)}^1 \mathbb{E}_{q_{t|0}(k|x_0)} \Big[ \sum_{j \neq k} Q(t)_{jk} \Big( s_\theta(k)_j - \frac{q_{t|0}(j|x_0)}{q_{t|0}(k|x_0)} \log s_\theta(k)_j + \psi\Big( \frac{q_{t|0}(j|x_0)}{q_{t|0}(k|x_0)} \Big) \Big) \Big] \mathrm{d}t, \quad (24)$$

where $\psi(y) \triangleq y \log y - y$. Note that our derivation above is different and simpler than that of Campbell et al. [29] (which Lou et al. [32] is based on) since we leverage the conditional reverse transition rate given $x_0$ instead of the transition rate matrix of the reverse process. We can further simplify the loss with the following relationship between the conditional score and $x_0$:

$$\frac{q_{t|0}(j|x_0)}{q_{t|0}(k|x_0)} = \frac{x_0^\top \bar{Q}(t) e_j}{x_0^\top \bar{Q}(t) e_k} = \frac{\alpha_t}{1 - \alpha_t} x_0^\top e_j \text{ for } k = m, j \neq k. \qquad (25)$$

Note that only the result under the case $k = m$ is needed. This is because when $x_t$ is unmasked, at any time between $0$ and $t$, the state must stay unchanged and remain $x_0$. As a result, $\mathrm{KL}(q(x_{t-\Delta t}|x_t, x_0) \| p_\theta(x_{t-\Delta t}|x_t)) = 0$ for $x_t \neq m$. From (15), we know $Q(t)_{jk} = \beta(t)(\delta_{mk} - \delta_{jk})$. Combining (25) and (24), we get

$$\mathcal{L}_\infty = \int_{t(1)}^1 \beta(t) \Big( \mathbb{E}_{q_{t|0}(k|x_0)} \big[ \delta_{mk} \big( \sum_{j \neq k} s_\theta(k)_j - \frac{\alpha_t}{1 - \alpha_t} x_0^\top \log s_\theta(k) \big) \big] + \psi\Big( \frac{\alpha_t}{1 - \alpha_t} \Big) \Big) \mathrm{d}t. \quad (26)$$

Further, we can show the connection between (26) and (4) by reverting the score parameterization to a mean parameterization using (23), or equivalently $s_\theta(x_t)_j = \frac{\alpha_t}{1 - \alpha_t} \mu_\theta(x_t)^\top e_j$. By doing so, we obtain

$$\mathcal{L}_\infty = \int_{t(1)}^1 \beta(t) \Big( \mathbb{E}_{q_{t|0}(k|x_0)} \big[ \delta_{mk} \big( \sum_{j \neq k} s_\theta(k)_j - \frac{\alpha_t}{1 - \alpha_t} x_0^\top \log \mu_\theta(k) \big] + \frac{\alpha_t}{1 - \alpha_t} \Big) \mathrm{d}t.$$

Observing that

$$\sum_{j \neq m} s_\theta(m)_j = \frac{\alpha_t}{1 - \alpha_t}, \qquad (27)$$

we conclude that this recovers the objective in (4). Interestingly, in Lou et al. [32] the score parameterization is not constrained to satisfy (27). That means the learned reverse model might be incompatible with the forward process.

Below, we prove Proposition 1 using the result from Eq. (25).

**Proof of Proposition 1**

$$\frac{q_t(j)}{q_t(m)} = \frac{\sum_{x_0} q_{t|0}(j|x_0) q(x_0)}{q_t(m)} = \frac{\sum_{x_0} q_{t|0}(j|x_0) q_{0|t}(x_0|m)}{q_{t|0}(m|x_0)} = \mathbb{E}_{x_0|x_t = m} \Big[ \frac{q_{t|0}(j|x_0)}{q_{t|0}(m|x_0)} \Big]$$

$$= \mathbb{E}_{x_0|x_t = m} \Big[ \frac{\alpha_t}{1 - \alpha_t} x_0^\top e_j \Big] = \frac{\alpha_t}{1 - \alpha_t} \mathbb{E}[x_0|x_t = m]^\top e_j.$$

$\square$

### H.4 Other related work.

**MaskGIT [39].** MaskGIT is a diffusion-inspired iterative denoising model for discrete image tokens obtained through models such as VQ-VAE [70]. Training of MaskGIT follows the steps: (a) Sample $t \in [0, 1]$. (b) Given a mask scheduling function $\gamma(t)$, sample $\gamma(t)N$ tokens to place masks. (c) For data $x_0$ of size $(m + 1) \times N$ and the partially masked state $x_t$, minimize the negative log-likelihood

$$\mathcal{L}_{\text{MaskGIT}} = -\int_0^1 \mathbb{E}_{x_t} \Big[ \sum_{n:x_t^{(n)}=m} (x_0^{(n)})^\top \log \mu_\theta^{(n)}(x_t, t) \Big] dt. \tag{28}$$

Our forward process satisfies $q_{t|0}(m|x_0) = 1 - \alpha_t$. Therefore, when we set the mask scheduling function as $\gamma(t) = 1 - \alpha_t$ we obtain a loss similar to (5) without the $\frac{\alpha_t'}{1-\alpha_t}$ weighting. Note that there remains a difference in the sampling distribution of $x_t$: in the masked diffusion forward process, tokens are sampled independently and do not necessarily yield exactly $(1 - \alpha_t)N$ mask tokens at time $t$, though the expected number is $(1 - \alpha_t)N$. One might be interested in whether the uniform weighting can be recovered by selecting an appropriate schedule $\alpha_t$. However, solving $\alpha_t$ such that $\alpha_t' = \alpha_t - 1$ yields $\alpha_t = ce^t + 1$ and there is no $c$ that satisfies both $\alpha_0 = 1$ and $\alpha_1 = 0$. This shows that training with the MaskGIT loss (28) may not be faithfully optimizing the model likelihood.

**Discrete flow matching [49].** For the linear schedule $\alpha_t = 1 - t$, our reverse transition rate matrix (8) conditioned on $x_0$ is:

$$R^{x_0}(t) = -\frac{\alpha_t'}{1 - \alpha_t} e_m (x_0 - e_m)^\top = \frac{1}{t} e_m (x_0 - e_m)^\top.$$

This is the same as the conditional reverse transition rate used in Campbell et al. [49, Eq. (22)]—note that their time $t$ is reversed, and the rate matrix was therefore in the form $R^{x_0}(t) = \frac{1}{1-t} e_m (x_0 - e_m)^\top$.

**SDDM [30].** Sun et al. [30] proposed a pseudo-likelihood-like objective for training discrete diffusion models that can also be applied to masked diffusion. However, their objective encounters the same challenge as Campbell et al. [29] — requiring $N$ passes of the mask prediction model. To mitigate this, they introduced a new transformer architecture, which unfortunately leads to some performance degradation.

**Blackout diffusion [50].** Santos et al. [50] proposed a "blackout" diffusion process that gradually diffuses images to a black state. While this approach is similar to masked diffusion on binary data, key differences emerge when dealing with larger state spaces. In their method, image pixel intensities gradually fade out, whereas ours directly transition to a mask state. Our method offers more flexibility, being applicable to general discrete state spaces without requiring predefined structural relationships. It also demonstrates competitive performance in image generation, achieving this without knowing pixel value proximity.

**SUNDAE [51, 71].** Unlike masked diffusion, SUNDAE uniformly corrupts data with random tokens in the vocab (known as uniform discrete diffusion [14]). Additionally, it uses a second loss term from cross entropy between clean data and 1-step unrolled model prediction. Similar ideas have been proposed in [72].

## I Details for state-dependent rates

### I.1 Derivations and time continuous limit

All derivations in this section assume that $x_t$ is a single token, while for $N$ tokens the masked diffusion with state-dependent rates factorises across the $N$ tokens. Learning from data of $N$ tokens using variational inference is discussed in App. I.2.

Given the forward transition $q(x_t|x_s)$ and marginal $q(x_s|x_0)$ derived in main text (Sec. 6) The reversal given $x_0$ is $q(x_s|x_t,x_0) = \mathrm{Cat}(x_s; \bar{R}^{x_0}(t,s)^\top x_t)$ for

$$\bar{R}^{x_0}(t,s)_{jk} = \begin{cases} \left(\frac{\alpha_s - \alpha_t}{1-\alpha_t}\right)^\top x_0 x_0^\top e_k & j = m, k \neq m \\ \left(\frac{1-\alpha_s}{1-\alpha_t}\right)^\top x_0 & j = m, k = m \\ \delta_{jk} & j \neq m. \end{cases}$$

or alternatively can be written as

$$q(x_s|x_t,x_0) = \frac{q(x_t|x_s)q(x_s|x_0)}{q(x_t|x_0)}$$

$$= \frac{\left[\frac{\alpha_t^\top x_s}{\alpha_s^\top x_s} x_s^\top x_t + (1 - \frac{\alpha_t^\top x_s}{\alpha_s^\top x_s})e_m^\top x_t\right]\left[\alpha_s^\top x_0 x_0^\top x_s + (1 - \alpha_s^\top x_0)e_m^\top x_s\right]}{\left[\alpha_t^\top x_0 x_0^\top x_t + (1 - \alpha_t^\top x_0)e_m^\top x_t\right]}. \tag{29}$$

To simplify this expression we consider the two cases: either $x_t = m$ (i.e. $x_t$ is mask) or $x_t \neq m$ where in the second case $x_t = x_0$. For the case $x_t = m$, the denominator in (29) simplifies as

$$q(x_t = m|x_0) = 1 - \alpha_t^\top x_0$$

due to $x_0^\top x_t = 0$ since $x_0 \neq m$, i.e. the observed token $x_0$ cannot be a mask. Then given that $x_t = m$ the probability that $x_s = x_t = m$ is

$$\frac{1 - \alpha_s^\top x_0}{1 - \alpha_t^\top x_0} = \frac{(\mathbf{1} - \alpha_s)^\top x_0}{(\mathbf{1} - \alpha_t)^\top x_0} = \left(\frac{\mathbf{1} - \alpha_s}{\mathbf{1} - \alpha_t}\right)^\top x_0 \tag{30}$$

while the remaining probability for $x_s = x_0 \neq m$ is

$$\frac{(\alpha_s - \alpha_t)^\top x_0}{1 - \alpha_t^\top x_0} = \frac{(\alpha_s - \alpha_t)^\top x_0}{(\mathbf{1} - \alpha_t)^\top x_0} = \left(\frac{\alpha_s - \alpha_t}{\mathbf{1} - \alpha_t}\right)^\top x_0. \tag{31}$$

Then, combining (30) and (31) to write $q(x_s|x_t = m, x_0)$ in an unified way yields the expression (11) in the main Sec. 6. In the second case, when $x_t = x_0 \neq m$, $q(x_s|x_t \neq m, x_0)$ from (29) simplifies dramatically and it becomes $q(x_s|x_t \neq m, x_0) = x_t^\top x_s$ which is a point mass that sets $x_s = x_t$.

**Derivation of the continuous-time limit of the loss in** (12). To simplify the notation, we let $\xi_{s,t} \triangleq \frac{\alpha_s - \alpha_t}{1 - \alpha_t}$. We first compute the KL divergence terms in the discrete-time ELBO as

$$\mathrm{KL}(q(x_s|x_t,x_0)\|p_\theta(x_s|x_t))$$

$$= \begin{cases} \sum_{x_s=0}^m q(x_s|x_t,x_0) \log \frac{q(x_s|x_t,x_0)}{p_\theta(x_s|x_t)} & x_t = m \\ 0 & x_t \neq m \end{cases}$$

$$= \delta_{x_t,m}\left[ \sum_{k \neq m} \xi_{s,t}^\top x_0 x_0^\top e_k \log \frac{\xi_{s,t}^\top x_0 x_0^\top e_k}{\xi_{s,t}^\top \mathrm{diag}(\mu_\theta(x_t,t))e_k} + (1 - \xi_{s,t})^\top x_0 \log \frac{(1 - \xi_{s,t})^\top x_0}{(1 - \xi_{s,t})^\top \mu_\theta(x_t,t)} \right]$$

$$= \delta_{x_t,m}\left[ -\xi_{s,t}^\top x_0 x_0^\top \log \mu_\theta(x_t,t) + (1 - \xi_{s,t})^\top x_0 \log \frac{(1 - \xi_{s,t})^\top x_0}{(1 - \xi_{s,t})^\top \mu_\theta(x_t,t)} \right].$$

Let $\Delta_t \triangleq \frac{1}{T} = t(i) - s(i)$ for all $i$. Plugging $\alpha_{t-\Delta t} = \alpha_t - \alpha_t' \Delta t + o(\Delta t)$ into the above formula and letting $\gamma_t = \frac{\alpha_t'}{1 - \alpha_t}$, we get

$$\mathrm{KL}(q(x_s|x_t,x_0)\|p_\theta(x_s|x_t))$$

$$= \delta_{x_t,m}\left[ \gamma_t^\top x_0 x_0^\top \log \mu_\theta(x_t,t)\Delta t + \left(1 + \gamma_t^\top x_0 \Delta t\right) \cdot \log \frac{1 + \gamma_t^\top x_0 \Delta t + o(\Delta t)}{1 + \gamma_t^\top \mu_\theta(x_t,t)\Delta t + o(\Delta t)} + o(\Delta t) \right]$$

$$= \delta_{x_t,m}\left[ \gamma_t^\top x_0 x_0^\top \log \mu_\theta(x_t,t)\Delta t + \left(1 + \gamma_t^\top x_0 \Delta t\right)\left(\gamma_t^\top x_0 \Delta t - \gamma_t^\top \mu_\theta(x_t,t)\Delta t + o(\Delta t)\right) + o(\Delta t) \right]$$

$$= \delta_{x_t,m}\left[ \gamma_t^\top x_0 x_0^\top \log \mu_\theta(x_t,t)\Delta t + \gamma_t^\top x_0 \Delta t - \gamma_t^\top \mu_\theta(x_t,t)\Delta t + o(\Delta t) \right]$$

$$= \delta_{x_t,m} \cdot \gamma_t^\top (x_0 x_0^\top \log \mu_\theta(x_t,t) + x_0 - \mu_\theta(x_t,t))\Delta t + o(\Delta t).$$

Therefore,

$$\lim_{T\to\infty}\sum_{i=2}^{T}\mathbb{E}_{q(x_{t(i)}|x_0)}[\mathrm{KL}(q(x_{s(i)}|x_{t(i)},x_0)\|p_\theta(x_{s(i)}|x_{t(i)})))]$$

$$=\lim_{T\to\infty}\sum_{i=2}^{T}\mathbb{E}_{q(x_{t(i)}|x_0)}[\delta_{x_{t(i)},m}\cdot\gamma_t^\top(x_0x_0^\top\log\mu_\theta(x_{t(i)},t(i))+x_0-\mu_\theta(x_{t(i)},t(i)))\Delta t+o(\Delta t)]$$

$$=\int_{t(1)}^{1}\gamma_t^\top\mathbb{E}_{q(x_{t(i)}|x_0)}[\delta_{x_t,m}\cdot(x_0x_0^\top\log\mu_\theta(x_t,t)+x_0-\mu_\theta(x_t,t))]dt.$$

Letting $t(1)\to 0$ proves the result.

## I.2 Training and gradient estimation

The model is applied to data consisted of $N$ tokens where $x_0=(x_0^1,\ldots,x_0^{(N)})$ and where each state in the masked diffusion is $x_t=(x_t^1,\ldots,x_t^{(N)})$. The reverse generated model has a factorizing transition conditional of the form $\prod_{n=1}^{N}p_\theta(x_s^{(n)}|x_t)$ where $p_\theta(x_s^{(n)}|x_t)=q(x_s^{(n)}|x_t^{(n)},\mu_\theta^{(n)}(x_t,t))$ has a form that depends on whether $x_t^{(n)}=m$ or $x_t^{(n)}\neq m$. For the first case:

$$p_\theta(x_s^{(n)}|x_t^{(n)}=m,\{x_t^{(k)}\}_{k\neq n})=\left(\frac{1-\alpha_s}{1-\alpha_t}\right)^\top\mu_\theta^{(n)}(x_t,t)e_m^\top x_s^{(n)}+\left(\frac{\alpha_s-\alpha_t}{1-\alpha_t}\right)^\top\mathrm{diag}(\mu_\theta^{(n)}(x_t,t))x_s^{(n)},$$

where $\mu_\theta^{(n)}(x_t,t)=\mathrm{softmax}(f_\theta(x_t))$ is a $m+1$ dimensional probability vector modelled by a NN (where the final value is constrained to be zero since $\mu_\theta^{(n)}(x_t,t)$ is a reconstruction of $x_0^{(n)}$ which cannot be mask, so in practice the NN classifier needs to have a softmax output only over the $m$ actual token classes). Crucially, note that the NN classifier receives as input the full state $x_t$ of all tokens, while additional time features to encode $t$ are also included. When $x_t^{(n)}\neq m$ the reverse transition model is set to be $p_\theta(x_s|x_t^{(n)}\neq m,\{x_t^{(k)}\}_{k\neq n})=(x_t^{(n)})^\top x_s^{(n)}$ which matches precisely $q(x_s^{(n)}|x_t^{(n)}=m,x_0^{(n)})=(x_t^{(n)})^\top x_s^{(n)}$ from the forward process.

The full negative lower bound for state-dependent rates and assuming $N$ tokens is given by

$$\mathcal{L}_\infty^{(N)}=\int_0^1\left(\frac{\alpha_t'}{1-\alpha_t}\right)^\top\mathbb{E}_{q(x_t|x_0)}\left[\sum_{n:x_t^{(n)}=m}(x_0^{(n)}-\mu_\theta^{(n)}(x_t,t)+x_0^{(n)}(x_0^{(n)})^\top\log\mu_\theta^{(n)}(x_t,t))\right]dt. \tag{32}$$

Given that each $\alpha_{t,i}=1-t^{w_i}$, the reverse model becomes

$$p_\theta(x_s^{(n)}|x_t^{(n)}\neq m,\{x_t^{(k)}\}_{k\neq n})=\left(e^{w\log\frac{s}{t}}\right)^\top\mu_\theta^{(n)}(x_t,t)e_m^\top x_s^{(n)}+\left(1-e^{w\log\frac{s}{t}}\right)^\top\mathrm{diag}(\mu_\theta^{(n)}(x_t,t))x_s^{(n)},$$

where $w$ is the $m+1$ dimensional vector of all $w_i$s. Note that the probability of $x_s^{(n)}$ staying in the mask state, i.e., $x_s^{(n)}=m$ depends on the full $x_t$ and it is given by $\left(e^{w\log\frac{s}{t}}\right)^\top\mu_\theta^{(n)}(x_t,t)=\sum_{i=0}^{m-1}e^{w_i\log\frac{s}{t}}\mu_\theta^{(n)}(x_t,t)_i$ while the probability for $x_s^{(n)}$ to take a certain non-mask token value $i$ is $\left(1-e^{w_i\log\frac{s}{t}}\right)\mu_\theta^{(n)}(x_t,t)_i$. The gradient wrt $t$ is $\alpha_{t,i}'=-w_it^{w_i-1}$ and $\frac{\alpha_{t,i}'}{1-\alpha_{t,i}}=-\frac{w_i}{t}$ the above loss is written as

$$\mathcal{L}_\infty^{(N)}=-\int_0^1\frac{1}{t}w^\top\mathbb{E}_{q(x_t|x_0)}\left[\sum_{n:x_t^{(n)}=m}(x_0^{(n)}-\mu_\theta^{(n)}(x_t,t)+x_0^{(n)}(x_0^{(n)})^\top\log\mu_\theta^{(n)}(x_t,t))\right]dt,$$

where $w$ is the vector of all $w_i$'s. An unbiased gradient over the NN parameters $\theta$ is straightforward to obtain since we just need to sample one time point $t$ and an $x_t\sim q(x_t|x_0)$ to approximate the integral and expectation and then use the gradient:

$$-\nabla_\theta\sum_{n:x_t^{(n)}=m}\frac{1}{t}w^\top\left(x_0^{(n)}-\mu_\theta^{(n)}(x_t,t)+x_0^{(n)}(x_0^{(n)})^\top\log\mu_\theta^{(n)}(x_t,t)\right).$$

The gradient wrt the $w$ parameters is more complex since these parameters appear also in the discrete distribution $q(x_t|x_0)$ which is not reparametrizable. To deal with this we need REINFORCE

unbiased gradients [73, 74], and in our implementation we consider REINFORCE leave-one-out (RLOO) [53, 54] with two samples. Firstly, the exact gradient wrt $w$ of the exact loss is written as

$$-\int_0^1 \frac{1}{t}\mathbb{E}_{q(x_t|x_0)}\left[g(x_t, x_0)\right] \mathrm{d}t - \int_0^1 \frac{1}{t}\mathbb{E}_{q(x_t|x_0)}\left[f(x_t, x_0)\nabla_w \log q(x_t|x_0)\right] \mathrm{d}t. \quad (33)$$

where

$$g(x_t, x_0) = \sum_{n:x_t^{(n)}=m} (x_0^{(n)} - \mu_\theta^{(n)}(x_t, t) + x_0^{(n)}(x_0^{(n)})^\top \log \mu_\theta^{(n)}(x_t, t)), \quad f(x_t, x_0) = w^\top g(x_t, x_0).$$

Note that $g(x_t, x_0)$ is a vector while $f(x_t, x_0)$ is a scalar. The left term in (33) is easy since it just requires sampling $t$ and $x_t \sim q(x_t|x_0)$, while the right term is the REINFORCE term which could have high variance. For this second term we use RLOO with two samples $x_t^1, x_t^2$ and construct the unbiased estimate

$$-\frac{1}{2t}\left(\nabla_w \log q(x_t^1|x_0) - \nabla_w \log q(x_t^2|x_0)\right)\left[f(x_t^1, x_0) - f(x_t^2, x_0)\right].$$

Thus, the overall unbiased gradient for $w$ we use is

$$-\frac{1}{2t}\left\{g(x_t^1, x_0) + g(x_t^2, x_0) + \left(\nabla_w \log q(x_t^1|x_0) - \nabla_w \log q(x_t^2|x_0)\right)\left[f(x_t^1, x_0) - f(x_t^2, x_0)\right]\right\}.$$

## J   Experimental Details

In all experiments, the model is trained with a continuous-time loss while samples are drawn from the discrete-time reverse model of 1000 timesteps unless otherwise noted. We used an exponential moving average factor 0.9999 for all evaluation including sample generation.

### J.1   text8

We followed the standard dataset split as in Austin et al. [14], Lou et al. [32] and trained our models on text chunks of length 256 for 1 million steps with batch size 512. All models in the table used a standard 12-layer transformer architecture unless otherwise noted. Our transformer has also the same number of heads (12) and hidden dimension (784) as in Austin et al. [14], Lou et al. [32].

We used the continuous-time ELBO and drew one sample of $t$ for each data to estimate the integral. To reduce the variance of training, we used the same antithetic sampling trick described in Kingma et al. [33] for continuous diffusion models. We used the linear masking schedule $\alpha_t = 1 - t$ and added a small shift $\epsilon = 10^{-4}$ when $t$ is close to 0 and 1 to ensure numerical stability. The shifted schedule is $\alpha_t = (1 - 2\epsilon)(1 - t) + \epsilon$. The shift leads to a support mismatch between $q(x_1|x_0)$ and the prior $p(x_1)$, leading to an undefined KL divergence term. We explain in app. E how to modify the prior distribution to allow small uniform probabilities in non-mask states to mitigate this problem. The shift leads to a non-zero reconstruction term and KL divergence term for the prior distribution but both are of negligible scale so we can safely ignore them when reporting the ELBO.

We used a cosine learning rate schedule with a linear warm up of 2000 steps. We applied channel-wise dropout of rate 0.05 and used AdamW optimizer with learning rate 0.0003 and a weight decay factor of 0.03. Our model is trained on 16 TPU-v5 lite for less than a day.

### J.2   OpenWebText

We kept 2% of the original training set for validation. Our small and medium transformer model have the same number of layers, heads, and hidden dimensions as in Lou et al. [32] and our tokenizer was also kept the same with a vocabulary size of around 50k. The training objective, masking schedule and other architectural choices were kept the same with the text8 experiment. We kept the training hyperparameters the same as text8 experiment except that we reduced the dropout rate to 0.02.

### J.3   FineWeb-Edu

We kept the same training setup as the OpenWebText experiments. Our transformer models have the same number of layers, heads, and hidden dimensions as those of GPT-2 models. We use the same GPT-2 tokenizer.

For Hellaswag evaluation, we concatenate question with each answer option, tokenize the concatenated string, pad to the length of 1024. The padded token sequence gets fed to our MD4 model's loss function for likelihood evaluation. We average 32 Monte Carlo samples to reduce variance. The answer with the highest likelihood estimate is the model's prediction.

### J.4  Images

We used the same linear masking schedule as in previous experiments in all likelihood results. We used the same U-Net plus self-attention architectures from the continuous diffusion model described in Kingma et al. [33] for CIFAR-10, except that we did not use Fourier feature inputs and added an additional input embedding layer with embedding size the same as the hidden dimension of the model. For ImageNet $64 \times 64$, we reduced the number of residual blocks (in one side of the U-Net structure) from 64 to 48 and added a 12-layer diffusion transformer [75] with 768 hidden dimension and 12 heads in the middle.

For both datasets we used AdamW optimizer and trained for 2M iterations. We used learning rate 0.0004, batch size 256, weight decay factor 0.01 for CIFAR-10 and learning rate 0.0002, batch size 512, weight decay factor 0.03 for ImageNet $64 \times 64$. The learning rate follows a cosine annealing after 100 warm up steps. Our CIFAR-10 model is trained on 32 TPU-v5 lite for 24 hours. Our ImageNet-$64 \times 64$ model is trained on 256 TPU-v5 lite for 3.5 days.

As explained in Sec. 4, we have observed that the cosine schedule leads to better sample quality so we used it to train a cheaper model for sample visualization. This model differs from the one that achieves best likelihood in that we used 8 residual blocks (in one side of the UNet structure) and a 20-layer diffusion transformer in the middle. All other configurations are kept the same.

## K  Additional Results

### K.1  Sample quality evaluation by GPT-2

We use the GPT-2 large model to evaluate the perplexity of samples generated by our model, following Lou et al. [32]. Results are shown in Fig. 8.

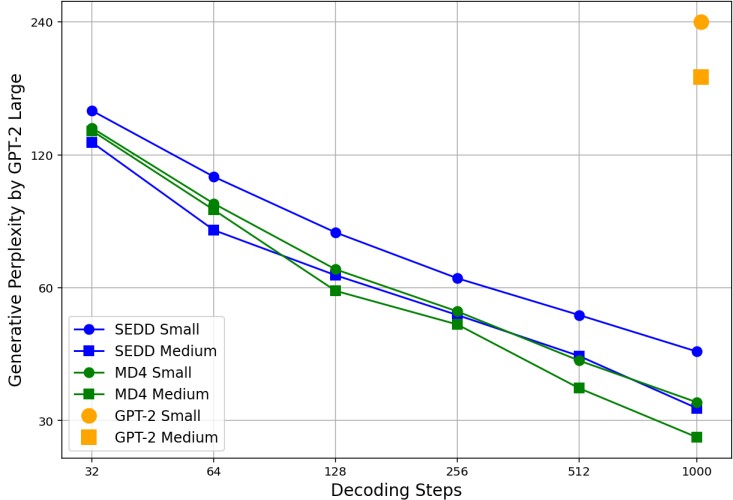

Figure 8: Generative perplexity evaluated by GPT-2 Large following Lou et al. [32]. We compare MD4 against the GPT-2 checkpoint (autoregressive baseline) and SEDD (the previous best discrete diffusion model on this task) in generating 1024-token text sequences. We investigate the effects of two orthogonal factors on sample quality: model size and decoding steps. The numbers for GPT-2 and SEDD are from Lou et al. [32].

### K.2 Perplexity on OpenWebText validation set

Tab. 5 reports the final perplexity number achieved on OpenWebText validation set, corresponding to Fig. 4.

Table 5: Perplexity on OpenWebText validation set.

| Size | Method | Perplexity ($\downarrow$) |
|---|---|---|
| Small | Gaussian Diffusion | $\leq 27.28$ |
| | SEDD Absorb (reimpl.) | $\leq 24.10$ |
| | MD4 (Ours) | $\leq 22.13$ |
| | GenMD4 (Ours) | $\leq \mathbf{21.80}$ |
| Medium | MD4 (Ours) | $\leq \mathbf{16.64}$ |

### K.3 FID evaluation of MD4 trained on ImageNet 64×64

We provide the FID numbers corresponding to Fig. 2 in Tab. 6.

Table 6: FID of 50k samples generated by MD4 trained on ImageNet $64\times 64$, corresponding to Fig. 2. Top three rows show results from an unconditional model, while the bottom row is from a model conditioned on class labels. Uniform discretization grid is used in Alg. 2 unless otherwise noted.

| Method | Timesteps $T$ | | | |
|---|---|---|---|---|
| | 64 | 128 | 256 | 512 |
| Linear $\alpha_t$ | 193.81 | 128.18 | 72.94 | 50.21 |
| Linear $\alpha_t$, cosine grid | **42.07** | 25.16 | 18.31 | **18.22** |
| Cosine $\alpha_t$ | 47.46 | **23.84** | **17.8** | 18.74 |
| Cosine $\alpha_t$, class conditional | **30.75** | **11.39** | **7.13** | **7.8** |

### K.4 Additional unconditional generation from MD4 trained on ImageNet 64×64

We provide more unconditional generation results from our pixel-level modeling experiments on ImageNet 64×64 in Fig. 9.

### K.5 Additional unconditional generation from MD4 trained on OpenWebText

Below we include two unconditioned text samples generated by our MD4 Medium model trained on OpenWebText.

#### K.5.1 MD4-M unconditional sample 1: 1024 tokens

```
like, I don't have to be alive? Sometimes there are things that are too real
and you're really supposed to experience them. So that's a good feeling.
That is the scary thing. Not actually, being able to experience things, being
able to do these things, when you're doing them, which, for most people
having to wake in a dream is something that seems the most significant, and then
you think about it the next day. It's like the hope of the future,
and you wake up right now thinking about it. What happens is,, then you
have to stop and think about it and then all of a sudden, somebody always
says, "You're dreaming."

And sometimes I wonder if this is a good time to teach your gut instincts to
your actors when you're doing a show like this. Because even on this particular
show, it feels like everyone's been through this all the time before, if even
a few years ago. I mean, if you're doing a show together, at least not on
continuous development, you you're a vet. I mean, you should really be along.
```

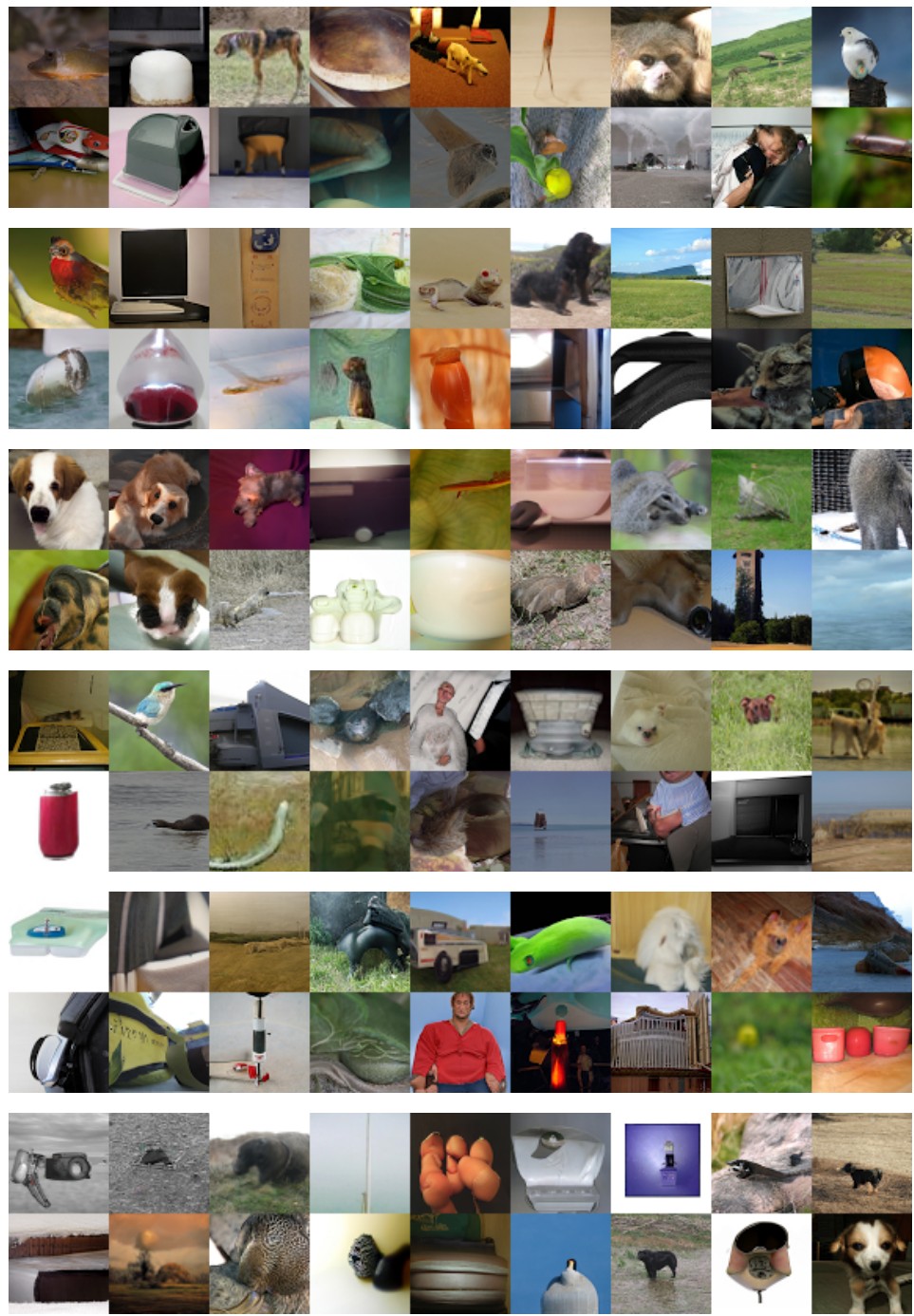

Figure 9: More unconditional samples from MD4 trained on ImageNet 64×64.

```
If you're not sure, well --

VS: I'm working on that one.

Did any of you guys feel that an instinct could work? I thought, "Well, because
you didn't do 'Deadwood' you should stop doing this." But when I read the story
for the first time, I thought, "I think this is going to work." What I can't
picture is a way to hold this apart.
```

VS: That's me. It's what we have to do. So do we. When we wrote the first episode,
we wrote a script that we felt like me and myself would want to see. I knew that I
wanted to be able to be in something -- and I wanted to be able to take refuge in
something that was real, that you could see and just really step out of yourself.
And then I saw it. Then, you get rehearsing it and doing it. And then I actually
started shooting. I think I knew I didn't think it was going to be good. But,
I know it was good. And now people are talked about because it's not good enough.

Growing up, you say that you just completely hated the show, "Lost." Isn't that
what you wish for at the end of the day?

VS: I don't like the concept.

And so there's a lot that you don't know about that, so I think for me to have had
these ideas, if you didn't understand even that it was coming out of this world
that doesn't exist, we might never get together.

It's so weird. This happened to happen at the same time?

VS: Yes. It happened to happen at basically the same time.

Nobody's even had a show or had a movie/come out of the movie, but ...

VS: If I'm going to pretend I'm definitely not you and have to live through that
stuff, I don't think I'm going to swallow that. I didn't expect it to do quite
that long.

There are always things now that happen with 'Deadwood' where you don't know where
it's going to end up next time, but I think there are occasions now where we have
to keep the fight, even if 'Lost' was pretty consistent in the mindset and the form.

VS: I'm glad that we did fight the odds, because we should have understood that
there was a direct link. But there was almost a sense of not that we had showed up
on the same day, we know we work in the same pieces, but a lot of stuff we don't
know about. Some of it, we need to deal with. We also just have to accept the
language, and there are a lot of things where we take from them and we do this
what they did  because we want to

### K.5.2   MD4-M unconditional sample 2: 1024 tokens

the groups let recreational vehicles use the three roads that will stay open in
the meantime of fighting off the permit. "The purpose of the permit is to make
sure that we work with  the NPS and made roadways and rest areas. We're not just
scaring guys kind of messing around." Community plans to build an urban bike
facility marched forward at the ongoing staff meeting of the King County
Commission.

Trail will be finished just south of the Greenview 5.

Instead of continuing with a pedestrian and bike trail to the MBTA's campus, these
two trails could bridle the areas from Market to 14 and carry communities closer.

"This project will provide a car-free path to King County," said Andrew Weed. It's
been put the brakes on in the past several months, but there are those residents
still skeptical.

"I've addressed some of the community concerns that've been raised. They've
expressed some of their concerns. I don't think it's terribly reasonable from a

transportation standpoint."

The trail had been set up to meet on for more than a year when the council approved funding for a different proposal.

Mayor Muriel Bowser said after meetings with Commissioner Bushell on Thursday that the new plan will be on board in December.

"There's enough of a finish for this project to roll out on time, and we're going to get it done," Bowser said.

For the public, the campaign appears over.

‘‘There was one meeting that I feel like I lost at last night's meeting," said Shelley Potts, a local resident.

Local resident Joel Grimy, who lives on Uman Road, met residents there as well.

And in other groups that rode through Mayor assistant Stacey Land and even her son held fliers saying to look for light sign, and also met with Bowser's son, Deion Bowser, about a future plan to also have a dog park on the transit corridor.

Advocates at Brickley's event, many one waited at least 11 minutes in during the start of the public meeting, said they expect at least another month from the Board of Commissioners, even after a public hearing on Nov. 13.

"We've been trying to be a talkative board where we are meeting in advance, being respectful of folks," Bowser said.

He considered that the proposal for the section of trail between the Greenview 5 and 3 ‘‘has to move on a schedule. We have other historic preservation projects that would take over that.’’

But Chad Routledge, a local advocate of the project, spoke out against the mayor's plan.

‘‘The mayor has sent a new meeting to the public using the same route that resulted from the loud criticism and onslaught of complaints from the community committee back during the public hearing,’’ Routledge said.

The BDC doesn't have a particular plan-turns around for the end of the planned path, and says ‘‘nothing practical can happen right now.’’ But, she said the agency still "looking to make investments in facilities along the route."

And still there is another part of the trail that might be just as much a wish for the dogs, as cars: the district wants to go west foot a couple blocks south, to make the trail safer for dogs.

‘‘I feel that the accessibility of the trail is pretty important. I think the education of the trail, and the uses along different routes are very important pieces of a balanced outcome,’’ said Bushell.

Trams coming off Route 1

### K.6   Conditional generation from MD4 trained on OpenWebText

We share conditionally generated text samples by MD4 Medium in Fig. 10 and observe that slow unmasking near $t = 1$, enabled by the cosine schedule, tends to help produce more consist and meaningful samples than uniform unmasking counterpart.

| MD4-M linear schedule | skydiving is a fun sport, but it's pretty risky. You're getting is one to get last one for the season if something goes wrong and it can happen you know, we know about season, especially in Skydiving, but anybody that wins this year | Then some time on Saturday you should pretty much say: "This is what I am going to be doing right now." It's just the simplest thing—that is why I always shampoo twice a day and shower three times a day. |
| MD4-M linear schedule + cosine grid | skydiving is a fun sport. It is pure endurance and excitement for many people in the at many times we could have won or lost. So if something goes wrong and we can't make it, we have to help another friend as if we have come to our zoo | First, just keep your skin as healthy as possible,you never want to be oily,that is why I always shampoo twice a day and shower three times a day. |
| MD4-M cosine schedule | skydiving is a fun sport, but it's extremely risky. You can have so many injuries one time and then one next time. There are so many ways you can hurt, so, neuroconcussions, especially from Skydiving, are continuing to rise every year | Though antibacterial products are a poison, the skin needs a chemical solution that protects it from bacteria and spots that form within it —that is why I always shampoo twice a day and shower three times a day. |

Figure 10: Conditionally generated text samples from MD4-M. Top: MD4-M trained with the linear schedule, sampled with a uniform grid; Middle: MD4-M trained with the linear schedule, sampled with the cosine grid; Bottom: MD4-M trained with the cosine schedule, sampled with a uniform grid. Context text shown in blue, model-generated text in black.

### K.7  Effect of discretization on zero-shot perplexity

We carried out ablation study on the effect of discretization on zero-shot perplexity. Results are included in Tab. 7. Note that this is an inference ablation with the same trained model (MD4-S trained with the continuou-time objective).

Table 7: Effect of discretization on zero-shot perplexity.

| Size | Timesteps | LAMBADA | WikiText2 | PTB | WikiText103 | IBW |
|------|-----------|---------|-----------|-----|-------------|-----|
| Small | T = 100 | $\leq 49.8$ | $\leq 36.1$ | $\leq 105.2$ | $\leq 36.1$ | $\leq 70.3$ |
| | T = 1000 | $\leq 48.5$ | $\leq 35.0$ | $\leq 102.5$ | $\leq 35.0$ | $\leq 68.4$ |
| | T = 10000 | $\leq 48.4$ | $\leq 34.9$ | $\leq 102.4$ | $\leq 34.9$ | $\leq 68.2$ |
| | T = $\infty$ (continuous) | $\leq 48.4$ | $\leq 34.9$ | $\leq 102.3$ | $\leq 35.9$ | $\leq 68.1$ |

