# OpenReview forum: "Simplified and Generalized Masked Diffusion for Discrete Data"
_NeurIPS.cc/2024/Conference — NeurIPS 2024 poster_

### Official Review · Reviewer_EzVa · 2024-07-07

**Soundness:** 4
**Presentation:** 4
**Contribution:** 4
**Rating:** 6
**Confidence:** 3

**Summary:**

This paper proposes a new framework for masked diffusion models for generative modeling of discrete data. Masked diffusion models offer an alternative to autoregressive models for discrete data but have faced challenges due to complex formulations and unclear relationships between different approaches. This paper presents a simplified and generalized framework to address these issues, enhancing the performance and training of masked diffusion models.

The key contributions includes:

1. Simplification of Model Formulation: The paper establishes properties for the forward process and its time reversal using elementary arguments, provides a simple expression for the Evidence Lower Bound (ELBO), demonstrating it as a weighted integral over time of cross-entropy losses, and shows invariance properties similar to continuous space diffusions.

2. Re-derivation of Training Objectives: The paper demonstrates how various previously proposed discrete diffusion training objectives can be derived from the ELBO objective by altering parameterization, relaxing constraints, or modifying loss weighting.

3. Performance Improvements: The paper demonstrates state-of-the-art likelihood and zero-shot transfer results on text and image tasks using the proposed ELBO objective.

4. Generalized Masked Diffusion Model: The paper proposes a generalized masked diffusion model that allows state-dependent masking schedules, further improving predictive performance on test likelihoods.

**Strengths:**

1. The paper makes a notable contribution to the field of generative modeling for discrete data by introducing a simplified and generalized framework for masked diffusion models.
2. The quality of the paper is reflected in the thoroughness of its methodology and the robustness of its experimental validation.
3. The paper is well-written and clearly structured, making it accessible to both experts and those new to the field.
4. The significance of the paper lies in its potential to impact a wide range of applications in generative modeling for discrete data.

**Weaknesses:**

1. While the paper provides a robust theoretical foundation, there could be more emphasis on practical applicability. The paper could benefit from additional practical guidelines for implementing the proposed framework, such as more detailed pseudocode and specific implementation challenges.
2. The experimental results presented are strong, but the range of tasks and datasets could be expanded, such as VQ-Diffusion [1] for token-based text-to-image.
3. I am unfamiliar with diffusion models for text generation. For image generation, the paper has reported likelihood results, missing some other common metrics, such as FID and IS.

[1] Vector Quantized Diffusion Model for Text-to-Image Synthesis

**Questions:**

Please see weaknesses.

**Limitations:**

The authors have adequately described the limitations and potential negative societal impact of their work.

---

> ### Author Rebuttal · Authors · 2024-08-07
>
> Thank you for the positive feedback! We are glad that you find our contribution notable, our methodology thorough and our experimental results strong and robust. We address each comment below
>
> ## Detailed pseudocode and specific implementation challenges
> We thank the reviewer for the suggestion. Please refer to our response to questions shared by reviewers for pseudocode of training and sampling algorithms. We will add them in the final version. We have discussed implementation challenges in Appendix A of the paper. Please let us know if you have any additional questions about the implementation.
>
> ## The range of tasks and datasets could be expanded, such as VQ-Diffusion [1] for token-based text-to-image.
> We thank the reviewer for the suggestion and will pursue this in future work. It's important to note that our pixel-level image modeling task is more challenging than modeling the latent space of vector quantized image encodings. While some discrete diffusion models have tackled high-resolution generation using the latter approach, to our knowledge, we're the first to go beyond 32x32 resolution for pixel-level modeling using discrete diffusion.
>
> ## Sample metrics for image generation
> In our rebuttal pdf we added an FID evaluation using 50K randomly generated samples from our MD4 models on ImageNet
> 64x64. We compared different alpha schedules (Linear v.s. Cosine) and show that the cosine schedule leads to significant better sample quality. We further trained a class-conditional model on ImageNet 64x64 that boosts the FID to around 7. Although these are not state-of-the-art FIDs on ImageNet 64x64, we emphasize our models are optimized for likelihood using ELBO instead of sample quality, unlike the reweighted objectives used in continuous diffusion.

---

> > ### Comment · Reviewer_EzVa · 2024-08-11
> >
> > I acknowledge having read the authors' rebuttal. My overall assessment of the paper remains unchanged, and I continue to support my current rating.

---

> > > ### Author Response · Authors · 2024-08-12
> > > **Response**
> > >
> > > Thank you for the acknowledgement and the positive assessment!

---

### Official Review · Reviewer_Pt68 · 2024-07-12

**Soundness:** 2
**Presentation:** 2
**Contribution:** 3
**Rating:** 6
**Confidence:** 5

**Summary:**

The paper simplifies the mathematical formula for the absorbing state diffusion process. By doing so, the authors derive a continuous-time ELBO for masked diffusion models. Their method, MD4, achieves better perplexity scores than SEDD on text8 and zero-shot perplexity on numerous datasets.

**Strengths:**

Simplifies the complex mathematical formulations for the absorbing state diffusion for D3PM.

**Weaknesses:**

Weaknesses:

1. Weak empirical results
    1. The zero shot numbers for D3PM in Table 1 look fishy.  There are only 2 differences between Md4 and Absorbing State D3PM:
        1. Mathematical simplification. In the discrete case (Eqn. 6), even though MD4 features a Simplified functional form for the ELBO, it shouldn't give it any performance benefits in terms of perplexity since it is mathematically equivalent to D3PM.
        2. The improvement in ELBO could be because of the continuous time formulation. However, VDM [1]  has shown that for gaussian diffusion, improvement from discrete (T=1000) to continuous time (T = $\infty$) barely improves the likelihood by less than 1%. For this reason, I request the authors to perform eval on an already trained model and report the  perplexity numbers on text8 or OWT using Eqn (6) with T=100, 1000, 10000. If the numbers reported for D3PM in Table 1 are indeed correct, and if the entire improvement is coming from the continuous time formulation, then the discrete time MD4 should get a number that's comparable to D3PM's zero shot ppl numbers.

        Questions: How did they retrain D3PM? Did they use the same transformer backbone as MD4? Did they use the same model size and data pre-processing scheme? Did they use  uniform state or absorbing state diffusion process? The authors need to clarify this.

    2. CIFAR10 Experiments. The AR baselines use old transformer models hence the comparison isn't quite fair. Current SOTA diffusion models on Imagenet 32 achieve a NLL of 2.55 [2] which is far better than the absorbing state diffusion models. So, I'm unsure about the takeaway from Table 3. In the conclusion section, the authors claim that "… on text and image data, the resulting masked diffusions outperform existing discrete and continuous diffusion models …" which is factually incorrect given that their method largely underperforms against gaussian diffusion [1, 2].
2. Limited evaluation of GenMD4. The authors mention that GenMD4 performs poorly on zero-shot tasks. I request the authors to quantify this poor performance by providing
    1. Validation ppl numbers on OWT
    2. zero-shot ppl numbers.

[1] Kingma, D., Salimans, T., Poole, B. and Ho, J., 2021. Variational diffusion models. *Advances in neural information processing systems*, *34*, pp.21696-21707.

[2] Sahoo, S., Gokaslan, A., Sa, C., Kuleshov, V., 2024.  Diffusion Models With Learned Adaptive Noise. arXiv:2406.07524

**Questions:**

1. Clarification on D3PM experiments in Table 1 as mentioned in the "weaknesses" section in the reviews.
2. Why did the authors decrease the dropout to 0.02 for OWT experiments and not set it to 0? Diffusion models are heavily regularized due to the randomness in the input to the model and oftentimes don't require additional regularization such as dropout. Hence, an intuitive or an empirical explanation would be helpful.

**Limitations:**

Yes

---

> ### Author Rebuttal · Authors · 2024-08-07
>
> Thank you for the time and the feedback! We clarify a few points.
>
> ##  D3PM results & differences between MD4 and D3PM
> * The reviewer is concerned if the D3PM results for zero-shot transfer tasks are comparable to MD4. We clarify that these D3PM results are from the SEDD paper (Lou et al., ICML 2024) which used the same setup as SEDD, including an absorbing state diffusion. This paper states:
> "... baselines are ... D3PM (with Absorbing Transition) … We retrain both models … reuse our model architecture and match hyperparameters (i.e. model size, training specifications)."
> To ensure direct comparability with MD4, we reimplemented SEDD and reproducing their results (see "SEDD Absorb (reimpl.)" in Table 1). This ensures a fair comparison between the models.
> * The reviewer states that MD4's simplified ELBO form shouldn't provide benefits compared to D3PM as both are ELBOs. However, we emphasize that this simplification is crucial for a well-engineered, numerically stable ELBO implementation, leading to significantly better performance. To illustrate, we analyze D3PM's original implementation of the $KL(q(x_s|x_t, x_0)||p(x_s|x_t))$ terms in the ELBO (s = t - 1 for D3PM) without simplification:
>
>     1. Sample $x_t \sim q(x_t|x_0)$
>     2. Calculate $q_{s|0}(\\cdot|x_0)$ where $\cdot$ takes all values in the vocabulary.
>     3. Calculate $q_{t|s}(x_t|\\cdot)$.
>     4. Calculate the logits of $q(x_s|x_t, x_0)$ via $qlogits = \\log (q_{s|0}(\\cdot|x_0) + \\epsilon) + \\log (q_{t|s}(x_t|\\cdot) + \\epsilon)$.
>     5. Repeat 2-4, replacing $x_0$ with NN output probabilities to get $p(x_s|x_t)$ logits ($plogits$).
>     6. Compute $KL(q(x_s|x_t, x_0)\\|p(x_s|x_t))$ using log_q = log_softmax(plogits) and log_p = log_softmax(qlogits).
>
> For masked diffusion, as shown by our simplification, many elements of $q(x_s|x_t, x_0)$ and $p(x_s|x_t)$ are zeros (see eq. (40) in App.). These terms can be removed but are instead included by D3PM. Thus, D3PM has to introduce an $\\epsilon$ to avoid inf - inf which gives NaNs in the KL computation. We perform an experiment in CIFAR10 to show the numerical issues: we replaced the MD4 objective in our code with D3PM’s ELBO and tried different $\\epsilon$s (1e-20, 1e-12, 1e-8, 1e-6). Only after increasing it to 1e-6 we avoided NaNs at the start of training. Still, we got NaNs after 300k iterations. The test BPD before NaNs was 3.03, while MD4 at this training iteration reports 2.89.
>
> * We agree with the reviewer that, similar to VDM, the cont-time v.s. discrete-time difference will not make a huge difference in likelihood. However, our cont-time formulation remains a key contribution for these reasons:
>     * It specifies the diffusion model using a simple monotonic function $\\alpha_t$ representing unmasking ratio at t. This enables exploration of schedules not used in discrete-time models like D3PM where the model is specified with less intuitive jump probabilities $\\beta_i$ rather than masking ratios. One of our key findings is that the cosine alpha_t schedule produces the best sample quality. In our rebuttal pdf, we include a comparison between the widely used linear schedule and the cosine schedule, showing the latter leads to significantly better FIDs in ImageNet 64x64.
>     * It enables easy adoption of training techniques from Gaussian diffusions; e.g. the antithetic sampling used by VDM for estimating the time integral in the ELBO which reduces training variance—on CIFAR-10, this translates to ~0.02 improvement on BPD.
>
>
> ## CIFAR10 comparison with AR
> The reviewer states that “The AR baselines use old transformer models hence the comparison isn't fair.” We have included the best AR results we found for CIFAR10. We're open to comparing with newer models if the reviewer can provide references. Also note our reported MD4 result of 2.78 is not fully optimized, e.g., increasing the batch size to 256 improves the result to 2.75 using the same 20M parameter model.
>
>
> ## CIFAR10 comparison with continuous diffusion
> The reviewer states that “Current SOTA diffusion models ... achieve a NLL of 2.55 [2] which is far better than absorbing diffusion” We believe the 2.55 from [2] (which we assume refers to CIFAR-10, not ImageNet 32) isn't directly comparable to our results in Table 3 for these reasons:
> * The 2.55 in [2] estimates exact log likelihood with probability flow ODE, while our results are upper bounds.
> * The best variational bound result in [2] is 2.65, which relies on learnable schedules using NNs, while MD4 uses a fixed schedule.
> * All continuous diffusion results in [2] Table 2 with BPD below 2.8 use extra image Fourier features (FFs) inputs from VDM. Comparing these with discrete diffusion is unfair since the latter assumes order-agnostic image data, i.e., the model is unaware of the proximity between pixels.
>
>
> Also the impact of FFs becomes negligible at larger scale such as ImageNet 64x64. We have found that MD4 on ImageNet 64x64, by reducing the dropout rate to 0.0, gives 3.40 BPD which is the same as VDM relying on FFs.
>
>
> We will revise the imprecise statement in the conclusion as  "On text data, our masked diffusions outperform existing discrete and continuous diffusion models. For pixel-level image modeling, we significantly improve discrete diffusion results, outperforming similar-sized AR models and achieving comparable likelihoods to continuous diffusion models without Fourier features."
>
>
> ## GenMD4 on zero-shot
> We clarify that we did not state GenMD4 performs poorly on zero-shot tasks. In fact, the results there are mixed. We have included both validation PPL numbers on OWT and zero-shot PPL numbers for GenMD4 in our response to shared questions by reviewers.
>
>
> ## Dropout rate for OWT
> Following the reviewer’s suggestion, we conducted dropout rate tuning and found that lower rate is better: dropout rate 0.05 has eval PPL 24.63, dropout rate=0.02, eval PPL is 22.13, and dropout rate = 0.0, eval PPL is 21.86. We’ll update the results in the final version.

---

> > ### Comment · Reviewer_Pt68 · 2024-08-11
> > **Update**
> >
> > >The AR baselines use old transformer models hence the comparison isn't fair.
> >
> > By newer models, I was referring to newer transformer architecture variations like the Llama 2 flavor of transformers or transformers, transformers with RoPE etc. The comparison to SEDD is fair comparison as noted, but it would be good to see more AR. Also non-AR non-sequence-based diffusion baselines should still be listed in Table 3 for the image models as the performance is nowhere near SOTA for non-sequence based non-AR models and it may confuse readers who are not familiar with generative models. Comparison to SOTA Gaussian diffusion models should be drawn, even though they are unfavorable to this class of the methods.
> >
> > I would encourage the author to study the numerical stability of their objective more in the final paper and it could make the submission much stronger.
> >
> > I have raised my score 6 to reflect that some of my concerns have been addressed.

---

> > > ### Author Response · Authors · 2024-08-12
> > > **Response**
> > >
> > > Dear Reviewer,
> > >
> > > Thank you for raising the score! We are glad that the rebuttal addressed some of your concerns.
> > >
> > > To address the other points you mentioned, could you clarify on the following points?
> > > * Llama2-style transformers with RoPE embeddings are dense models. When applied to images (CIFAR-10 has a sequence length of 3072), they can be significantly more computationally expensive than the models in Table 3. All AR models in Table 3 are sparse or low-rank, while the diffusion models use UNet architectures rather than transformers to manage computational costs. Are you suggesting we compare to dense AR models? We believe such a comparison would be unfair in terms of computational requirements.
> > > * By non-AR non-sequence-based diffusion baselines, are you referring to continuous diffusion models?
> > >
> > > Best,
> > > Authors

---

### Official Review · Reviewer_Y9JA · 2024-07-12

**Soundness:** 3
**Presentation:** 3
**Contribution:** 3
**Rating:** 7
**Confidence:** 4

**Summary:**

The paper proposes a streamlined and generalized framework for masked diffusion models, addressing the complexities and inefficiencies of existing models, including those based on Score Entropy Discrete Diffusion (SEDD). It introduces a continuous-time variational objective for masked diffusion models, simplifying the evidence lower bound (ELBO) to a weighted integral of cross-entropy losses. Additionally, the paper presents state-dependent masking schedules, enhancing the flexibility and performance of these models. The proposed methods demonstrate state-of-the-art results in text and image tasks, significantly improving likelihood and zero-shot transfer performance.

**Strengths:**

- The paper offers a novel theoretical formulation of the continuous-time variational objective for masked diffusion models, simplifying the training process and ensuring consistency between forward and reverse processes.
- The introduction of state-dependent masking schedules provides a more adaptable approach, catering to the specific characteristics of the data and improving model performance.
- The proposed methods achieve state-of-the-art performance in both text and image generative tasks, significantly enhancing likelihood and zero-shot transfer capabilities.
- By reducing the ELBO to a weighted integral of cross-entropy losses, the paper makes the training and understanding of masked diffusion models more accessible and potentially more stable.
- The paper includes comprehensive experimental validation on various datasets, demonstrating the robustness and superiority of the proposed methods.

**Weaknesses:**

-  Despite the theoretical simplifications, the practical implementation of state-dependent masking schedules can still be complex and computationally demanding. Specifically, obtaining the starting x_T is challenging, and since the sampling process lacks stochasticity, sampling cannot be done from the completely masked state.
- The state-dependent models have a tendency to overfit to dataset statistics, which can limit their effectiveness in zero-shot transfer tasks.
- While the paper demonstrates superior performance, a more detailed comparative analysis with other state-of-the-art methods, particularly regarding computational efficiency and training times, would provide a clearer picture of the advantages.

**Questions:**

- Could the authors provide more insights into the practical challenges faced during the implementation of the state-dependent masking schedules?
- How does the proposed model ensure consistency between the forward and reverse processes, and how does this impact training stability compared to SEDD?
- Could the authors provide a detailed and separate description of the training and sampling algorithms, similar to what is provided in the Appendix of the SEDD paper, to better and more easily understand the proposed method?
- How sensitive is the proposed method to hyperparameter choices? Do multiple runs with the same hyperparameters yield consistent performance?

---

> ### Author Rebuttal · Authors · 2024-08-07
>
> Thank you for the time you’ve taken to review our work and for the positive and constructive feedback! We are glad that you find our work "offers a novel theoretical formulation" and "achieve state-of-the-art performance" with " comprehensive experimental validation". We address each individual comment below.
>
> ## Practical challenges in implementation of state-dependent masking schedules
> * Regarding computational cost, the state-dependent masking schedules and corresponding GenMD4 objective require only twice the computation of MD4 with the same network size. While slightly more expensive, this increase remains within affordable limits.
> * Regarding sampling, we clarify that x_T is a full mask state even for the state-dependent schedule case. This is because the learned schedule $\\alpha_t = 1 - t^w$ still satisfies $\\alpha_1 = 0$ regardless of the value of $w>0$. We also clarify that the sampling process is stochastic in each step, please refer to the sampling algorithm described in our response to questions shared by reviewers.
>
>
> ## Effectiveness of state-dependent models in zero-shot transfer tasks
> We have included the results of GenMD4 on zero-shot transfer tasks in our response to shared questions. The results are mixed: on some datasets (Lambada, PTB) GenMD4 results in significantly better zero-shot PPL numbers than MD4 while on other datasets (WikiText) GenMD4 is slightly worse. We believe this indicates some degree of overfitting to the dataset statistics of OpenWebText.
>
>
> ## Comparative analysis regarding computational efficiency and training times
> We follow the reviewer’s suggestion to compare the training times of MD4/GenMD4 on OpenWebText using 128 TPUv3. We can see that MD4 is as fast as our reimplementation of SEDD while achieving better results. GenMD4 is slightly slower than MD4 due to the 2x computational cost.
> * Gaussian Diffusion: 3.5 steps / s
> * SEDD: 4.2 steps / s
> * MD4: 4.2 steps / s
> * GenMD4: 3.5 steps / s
>
>
> ## How the model ensures forward/reverse consistency and impact on training stability v.s. SEDD
> For simplicity let’s consider the single-dimension case. As we have shown in Proposition 1, the true score function for a mask state $x_t=m$ has the following relationship with the conditional mean of $x_0$:
> $$
> s(m, t)\_j = \\frac{\\alpha_t}{1 - \\alpha_t} E[x_{0,j}|x_t=m]
> $$
> Note that here $x_0$ is a one-hot representation of the data therefore $\\sum_j x_{0,j} = 1$ and thus also $\\sum_j E[x_{0,j}|x_t=m] = 1$. This implies that
> $$
> \\sum_j s(m, t)_j  = \\frac{\\alpha_t}{1 - \\alpha_t}.
> $$
> Since the transition rate matrix of the true reverse process depends on the score, the above equation implies a constraint that the reverse process has to satisfy in order to be consistent with the $\\alpha_t$-determined forward process.
>
>
>
>
> As we have explained in Sec. 4 and App. F3, we can interpret both our method and SEDD as learning a model $s_{\\theta}$ for the true score function $s$. The differences are
> * In MD4, $s_{\\theta}$ is parameterized as
> $$
> s_{\\theta}(m, t)\_j = \\frac{\\alpha_t}{1 - \\alpha_t} \\mu_{\\theta}(m)\_j
> $$
> where $\\mu_{\\theta}$ has a softmax output that produces a probability vector. Therefore, the score model also satisfies $\\sum_j s_{\\theta}(m, t)\_j = \\frac{\\alpha_t}{1 - \\alpha_t}\\sum_j \\mu_{\\theta}(m)\_j  = \\frac{\\alpha_t}{1 - \\alpha_t}$.
> * In SEDD, $s_{\\theta}$ is a free-form neural network model that outputs a real-valued vector, which has no guarantee it satisfies the constraint. This inconsistency between forward and reverse process leads to large variational gaps in the ELBO which we have empirically shown to lead to unstable training (see Figure 3).
>
>
> ## Detailed description of the training and sampling algorithms
> Please refer to our response above to questions shared by reviewers.
>
>
> ## Sensitivity to hyperparameter choices and multiple runs
> We test the sensitivity to multiple runs by re-training MD4-S with the same hyperparameter as the original one reported in the paper, where the original PPL on OpenWebText eval split is 22.126 and the re-trained MD4-S gets 22.134. In the pdf we uploaded for rebuttal, we include an analysis of the impact of masking schedule choice on sample quality, showing the benefits of a cosine schedule.

---

### Official Review · Reviewer_bny3 · 2024-07-12

**Soundness:** 3
**Presentation:** 4
**Contribution:** 3
**Rating:** 7
**Confidence:** 5

**Summary:**

Summary: This paper introduces a framework for masked diffusions that consolidates previous research on the topic and organizes it into a cohesive structure. The authors also present a generalized model within this framework, which enables the use of state-dependent masking schedules and optimization of scheduler parameters.

**Strengths:**

1. The GenMD4 framework offers a valuable approach to optimize the forward process. In earlier studies, forward processes were typically manually designed and set within the model. However, GenMD4 adjusts the forward distribution to align with the estimated distribution, thereby improving the forward process. This innovation may serve as a source of inspiration for developing more effective forward processes.

2. This paper summarizes previous formulations of masked diffusion models and establishes the connections between them.

**Weaknesses:**

1. In line 90. The handling of $p(x _0|x _{t(1)})$ could be enhanced. Assuming $p(x _0|x _{t(1)}) \propto q(x _{t(1)} | x _0)$ is equivalent to assuming that $q(x _0)$ is uniformly distributed. In reality, it should be treated the same as other $p(x _s|x _t)$.
2. In line 114. When discussing multidimensional data, it is not straightforward to assume that the backward process factorizes across tokens. This is because the distribution $p(x _0)$ does not factorize across tokens. Achieving factorization necessitates a small time step dt, which may not be easily observable. Additionally, in the previous single-token scenario, dt dose not need to be small, indicating that one step is sufficient to model the distribution $p(x _0 | x _1)$. This aspect is crucial for multidimensional data and should be emphasized in a fundamental paper like this.
3. In append F. The presence of a non-zero $\alpha _1$ may result in the "medium brightness problem" [1]. However, there is no singularity when $\alpha _1$ is zero if log-SNR is not introduced, and the time interval can be extended to [0, 1].
4. In append G2. When applied to masked diffusion, $R_{kj}$ is zero when $ j \ne k$ and $j \ne m$. Given that $R_{kk} + R_{km} = 0$, $\tilde{q}$ can only take on one value (m), resulting in no additional variance.
5. In image experiments, MD4 employs masked noise, while $\tau$LDR uses Gaussian noise. We recommend conducting experiments with the same noise scheduler to demonstrate conclusively that MD4 is superior. If the goal of this paper is solely to establish that masked noise outperforms Gaussian noise, we recommend explicitly stating this claim. Additionally, we advise detailing the sampling method, as variations in methodology can influence the quality of generated samples.

[1] Common Diffusion Noise Schedules and Sample Steps are Flawed, Lin et al., 2024

**Questions:**

1. GenMD4 has not been tested on image datasets. Could you please share the results of GenMD4 when applied to image datasets?
2. Since introducing GenMD4 results in additional variance, what if all tokens share the same w (referred to as "simplified-GenMD4")? This would result in less variance. Given that GenMD4's performance is close to MD4, can simplified-GenMD4 achieve the same BPC?

**Limitations:**

The method is only applied to masked diffusions.

---

> ### Author Rebuttal · Authors · 2024-08-07
>
> Thank you for the time you’ve taken to review our work and for the positive and constructive feedback! We are glad that you found our paper "offers a valuable approach to optimize the forward process”, “organizes prior work into a cohesive structure”, and "may serve as a source of inspiration" for future work. We respond to each of your comments below.
>
>
> ## The handling of 𝑝(𝑥0|𝑥𝑡(1)) could be enhanced
> Thank you for the suggestion. In our implementation t(1) is tiny ($\\alpha_{t(1)} = 10^{-4}$) and we expect p(x0|xt(1)) to be very close to q(xt(1)|x0). We have also tried treating p(x0|xt(1)) the same as other p(xs|xt) and did not observe significant differences.
>
>
> ## Multidimensional data and backward factorization
> Thank you for the suggestion. We agree with the reviewer that the true backward distribution becomes factorized only as $s \\to t$, which is why we adopt the continuous-time formulation throughout the work. We will follow the reviewer’s suggestion to make this more clear in the final version.
>
>
> ## Non-zero $\\alpha_1$
> We originally introduced the non-zero $\\alpha_1$ to resolve numerical issues in GenMD4, where the REINFORCE LOO gradient requires calculating $\\log \\alpha_t$. We have since then used the same schedule for MD4 as well. We agree with the reviewer that the non-zero $\\alpha_1$ is unnecessary for MD4. Although we have not found any “medium brightness” problems, we will still rerun the MD4 experiment with $\\alpha_1 = 0$ and update the results.
>
>
> ## Appendix G2: Variance of Campbell et al. (2022)
> We thank the reviewer for noting this. In the single dimension case, we agree that there is no additional variance via further simplification of the Campbell et al. bound. The additional variance comes from multidimensional data where the sum over $j$ becomes the sum over all neighbors in the forward process, i.e., the states that mask out a single dimension of $x_t$ that has not been masked yet. In this case, the Campbell et al. (2022) bound simplifies to
> $$
> \\beta(t) \\sum_{i: x_{t, i} \\neq m} \\log (x_{t, i}^\\top \\mu(x_t(i\\to m))_d
> $$
> where $x_t(i\\to m)$ denotes the state we get by masking out the i-th dimension of $x_t$. This equation requires $d$ function evaluations of the prediction model $\\mu$, thereby incurring additional variance if we estimate the sum with Monte Carlo. We will fix the discussion of Campbell et al. (2022) in the final version.
>
>
> ## 𝜏LDR with masked noise
> We followed the reviewer’s suggestion to perform an additional experiment using 𝜏LDR with masked noise. We made sure the neural networks and training hyperparameters used by 𝜏LDR (masked) is the same with our MD4 experiment. The results are as follows (for comparison we also included the MD4 results)
>
>
> | Method | BPD (CIFAR-10) |
> | :------ | ------: |
> | 𝜏LDR (masked) | <= 3.52 |
> | MD4 | <=2.78 |
>
>
> The results show that 𝜏LDR with masked noise suffers from the high-variance objective even with the same networks and training hyperparameters from MD4.
>
>
> ## Details of the sampling method
> Please refer to our response to questions shared by reviewers.
>
>
> ## Results of GenMD4 on image datasets
> We ran GenMD4 on both image datasets with the same architecture and hyperparameters as in MD4. On CIFAR-10, we get 2.7749 (GenMD4) v.s. 2.7847 (MD4). On ImageNet 64x64, we get 3.4233 (GenMD4) v.s. 3.4273 (MD4). We see consistent improvements in test BPD over MD4 throughout training although the improvement is smaller than that of text experiments (1.34 v.s. 1.37 BPC on text8 & 21.80 v.s. 22.13 PPL on OpenWebText validation set).
>
>
> ## Simplified GenMD4
> We followed the reviewer’s suggestion to perform an additional experiment on text8 that only learns a scalar w for GenMD4 that is shared across all schedules. The result is 1.37 BPD which is almost the same as the result of MD4 and worse than GenMD4 with a vector w (1.34 BPD). We believe this is because the simplified GenMD4 reduces to MD4 with a learnable schedule (we can show this by choosing $\\alpha_t$ in (19) as a scalar schedule times an all-one vector) and MD4 is invariant to masking schedules.

---

> > ### Author Response · Authors · 2024-08-09
> > **Correction for Typos**
> >
> > Dear Reviewer,
> >
> > We noticed a few typos in the original rebuttal and would like to correct them:
> > * It should be  $\\alpha_{t(1)} = 1 - 10^{-4}$ in the response to "The handling of 𝑝(𝑥0|𝑥𝑡(1)) could be enhanced"
> > * The equation  in the response to "Appendix G2: Variance of Campbell et al. (2022)" should be
> > $$
> > \beta(t) \sum_{i: x_{t, i} \neq m} \log (x_{t, i}^\top \mu(x_t(i\to m))_i)
> > $$

---

> > > ### Comment · Reviewer_bny3 · 2024-08-12
> > >
> > > The authors' rebuttal addresses my main concern, and I will adjust my score accordingly.
> > >
> > > However, there are still areas for improvement.
> > >
> > > 1. In Appendix G2, the reduction in variance is noticed, but the underlying reason deserves further elaboration. The masked diffusion is an instance of a CTMT diffusion where the reverse transition rate $\hat{R}_ {kk}$ remains constant, resulting in the omission of $\hat{R}_ {kk}$ in the ELBO. This strategic approach could serve as a valuable technique for designing discrete diffusion models.
> > >
> > > 2. the paper solely provides proofs and derivations in one dimension, yet there are significant distinctions between one dimension and multidimensional scenarios. It is recommended that the authors include proofs and derivations for the multidimensional case in the final version of the paper.

---

> > > > ### Author Response · Authors · 2024-08-13
> > > > **Response**
> > > >
> > > > Dear Reviewer,
> > > >
> > > > We're pleased our rebuttal addressed your main concern and appreciate the increased score. We'll incorporate your suggestion to add a detailed multidimensional case derivation in the final version. Regarding your follow-up question:
> > > >
> > > > > In Appendix G2, the reduction in variance is noticed, but the underlying reason deserves further elaboration.
> > > >
> > > > We believe the primary reason for variance reduction is the discrete "integration-by-parts" that transformed equation (54) (Campbell et al. form) to (55) (which simplifies to our loss). This rewrite mainly changes $R_{\\theta}(t)\_{jk}$ to $R_{\\theta}(t)\_{kj}$ in the second term. Since the first index of $R_{\\theta}(t)$ is the network input and we sum over $j$, having $j$ on the second index avoids evaluating the network at different inputs. As you correctly pointed out, this issue only arises for multidimensional data, since in the single-dimension case there's only one possible $j \neq k$.  In the final version we'll revise this section using multidimensional derivations for clarity.
> > > >
> > > > Best,
> > > > Authors

---

> > > > > ### Comment · Reviewer_bny3 · 2024-08-13
> > > > >
> > > > > Perhaps we consider different terms. Utilizing the discrete "integration-by-parts" technique can reduce the variance of the second term.
> > > > >
> > > > >  It is worth noting that due to the specific structure of the generator of masked diffusion, the term $R^{\theta}_ {kk}$ remains constant and does not depend on $\theta$, resulting in a variance reduction of the first term to zero. This unique feature is distinct from general discrete diffusion models. Exploring alternative classes of discrete diffusion models exhibiting this property presents a significant and meaningful research topic.
> > > > >
> > > > > I suspect that equation (7) is equivalent to the second term in equation (55), which could potentially unveil a deeper connection between the two equations.
> > > > >
> > > > > Bests

---

> > > > > > ### Author Response · Authors · 2024-08-14
> > > > > > **Response**
> > > > > >
> > > > > > Thank you for the clarification. We agree that for masked diffusion, the first term becomes constant, meaning equation (7) is equivalent to the second term of equation (55). It's important to note that the transition rate matrix satisfies $R_{kk} = - \\sum_{j\neq k} R_{kj}$, which can be combined with the second term of equation (55). This further demonstrates the advantage of equation (55) over equation (54).

---

### Author Rebuttal · Authors · 2024-08-07

# Response to comments shared by reviewers:

We thank the reviewers for their feedback. Below we address the questions shared by reviewers. We also uploaded a rebuttal pdf that contains figures used to address individual questions/comments of the reviewers.

## bny3,Y9JA: Details of training and sampling algorithms
A single step of the MD4 training algorithm is described below:


Input: data batch $x_0$, network $\\mu_{\\theta}(\\cdot, t)$, masking schedule $\\alpha_t$ \
Draw a batch of $t$ from U[0, 1] using antithetic sampling \
Sample $x_t$ using $q(x_t|x_0)$ by masking out independently each dimension of $x_0$ with probability $1 - \\alpha_t$ \
Get prediction logits via $\\mu_{\\theta}(x_t, t)$ \
Calculate the cross entropy loss $CE$ using $x_0$ and the prediction logits and sum over the masked dimensions of $x_t$. \
Compute unbiased estimate of the negative ELBO as $-\\frac{\\alpha_t’}{1 - \\alpha_t} * CE$  \
Optimize the negative ELBO via autodiff.


Throughout the paper we simply run ancestral sampling from a discrete-time backward model. Specifically, the sampling algorithm is:


Input: Context size N, discretization grid $0 = t(0) < t(1) < \\cdots < t(T) =1$ \
Init: $x_1 \\gets [m, \\dots, m]$ \
for $i=T, T - 1, \\dots, 1$: \
    $t \\gets t(i)$, $s \\gets t(i-1)$ \
    for $n \\in [N]$, if $x_t^{(n)} = m$, draw $x_s^{(n)} \\sim \\mathrm{Cat}(\\frac{\\alpha_s - \\alpha_t}{1 - \\alpha_t} \\mu_\\theta^{(n)}(x_t, t) + \\frac{1- \\alpha_s}{1 - \\alpha_t} e_m)$ else $x_s^{(n)} \\gets x_t^{(n)}$ \
return $x_0$


We will follow the reviewer’s suggestion to include both algorithms in the final version.


## Y9JA, Pt68: Evaluation of GenMD4 on OpenWebText validation set and zero-shot tasks
* GenMD4 leads to better perplexity on OpenWebText validation set (this result is already shown in Figure 3 of the submission, here we report the precise values in a table):




| Method                | Perplexity |
|-----------------------|-----------------------|
| Gaussian Diffusion    | <= 27.28              |
| SEDD Absorb (reimpl.) | <= 24.10              |
| MD4 (Ours)            | <= 22.13              |
| GenMD4 (Ours)         | <= **21.80**          |




* For zero-shot results, on some datasets (Lambada, PTB) GenMD4 results in significantly better zero-shot PPL numbers than MD4 while on other datasets (WikiText) GenMD4 is slightly worse. We believe this indicates some degree of overfitting to the dataset statistics of OpenWebText.


| Method | LAMBADA | WikiText2 | PTB | WikiText103 | IBW
| :------ | ------: | ------: | ------: | ------: | ------: |
| MD4 (Ours) | 48.43 | **34.94** | 102.26 | **35.90** | 68.10 |
| GenMD4 (Ours) | **33.31** | 41 | **65.06** | 41 | **52.1** |

---

### Decision · Program_Chairs · 2024-09-25

**Decision:**

Accept (poster)

**Comment:**

Overall, reviewers agree that the proposed framework for diffusion-based generative modeling of discrete data provides interesting conceptual advances, and leads to promising experimental results.  Reviews and author rebuttals highlight some areas where conceptual and experimental comparisons to prior work could be improved, please be sure to incorporate these points in the final version of your manuscript.